# Asymmetric chromatin retention and nuclear envelopes separate chromosomes in fused cells in vivo

Bharath Sunchu[1,3], Nicole MynYi Lee [1,4], Jennifer A. Taylor [1,5], Roberto Carlos Segura [1,5], Chantal Roubinet [2] & Clemens Cabernard [1✉]

Hybrid cells derived through fertilization or somatic cell fusion recognize and separate chromosomes of different origins. The underlying mechanisms are unknown but could prevent aneuploidy and tumor formation. Here, we acutely induce fusion between *Drosophila* neural stem cells (neuroblasts; NBs) and differentiating ganglion mother cells (GMCs) in vivo to define how epigenetically distinct chromatin is recognized and segregated. We find that NB-GMC hybrid cells align both endogenous (neuroblast-origin) and ectopic (GMC-origin) chromosomes at the metaphase plate through centrosome derived dual-spindles. Physical separation of endogenous and ectopic chromatin is achieved through asymmetric, microtubule-dependent chromatin retention in interphase and physical boundaries imposed by nuclear envelopes. The chromatin separation mechanisms described here could apply to the first zygotic division in insects, arthropods, and vertebrates or potentially inform biased chromatid segregation in stem cells.

[1] Department of Biology, University of Washington, Seattle, WA, USA. [2] Medical Research Council Laboratory of Molecular Biology, University of Cambridge, Cambridge, UK. [3]Present address: Department of Biology, University of Virginia, Charlottesville, VA, USA. [4]Present address: Department of Neuroscience and Behavioural Disorders, Duke-NUS Graduate Medical School, Singapore, Singapore. [5]These authors contributed equally: Jennifer A. Taylor, Roberto Carlos Segura. ✉email: ccabern@uw.edu

Dividing cells equally distribute the replicated chromosomes between the two sibling cells through microtubule-dependent attachment and segregation mechanisms[1,2]. Microtubules of bipolar spindles are connected to chromosomes via kinetochore proteins, which are localized on centromeric DNA[3,4]. Mitotic metazoan cells usually only form a single bipolar spindle, but certain insect species, arthropods, or mouse zygotes form two distinct mitotic spindles (dual-spindles, hereafter) in the first division after fertilization, which physically separates the maternal from the paternal chromatin[5–7]. Chromosome separation also occurs in hybrid cells, derived from somatic cell–cell fusion events[8,9]. Dual-spindle dependent chromosome separation could be achieved through specific chromosome recognition mechanisms or physical boundaries to distinguish and separate epigenetically distinct chromatin. The molecular nature of potential recognition mechanisms is not known but could entail asymmetries in centromere binding proteins, kinetochore size or kinetochore composition[10–13].

Over a century ago, unregulated cell-cell fusion between different somatic cells was proposed to initiate tumor formation[14]. Aichel's cell fusion model has the advantage that it can readily explain aneuploidy, a feature frequently observed at the early stages of tumor development[15,16]. Tetraploidy and supernumerary centrosomes—the natural products of cell fusion—predispose cells to aneuploidy through chromosome rearrangements[17]. Aichel's cell fusion model still remains to be experimentally validated, which requires a detailed characterization of chromosome dynamics in fused cells in vivo.

Here, we ask how hybrid cells derived through cell–cell fusion of molecularly distinct cell types accurately recognize, separate, and segregate epigenetically distinct chromosomes. To this end, we acutely fused *Drosophila* neural stem cells (neuroblasts (NBs)) with differentiating ganglion mother cells (GMCs) in the intact larval fly brain to create hybrid cells containing both neuroblast and GMC chromosomes. In contrast to previously reported cell–cell fusion experiments, performed by fusing cultured cells of the same type[8,9], we aimed to generate hybrid cells in vivo between molecularly distinct cell types.

Unperturbed *Drosophila* neuroblasts divide asymmetrically, self-renewing the neural stem cell while forming a differentiating GMC. Neuroblasts are twice the size of GMCs, express the transcription factor Deadpan (Dpn$^+$) and divide asymmetrically with a rapid cell cycle. The smaller GMCs can also be identified based on Prospero (Pros$^+$) expression and divide only once with a long cell cycle[18]. NB and GMC chromatin is epigenetically distinct, manifested in Histone modification differences[19]. Neuroblasts are intrinsically polarized, consisting of an apically localized Par complex, which is connected to the Pins complex, composed of Partner of Inscuteable (Pins; LGN/AGS3 in vertebrates), Gαi and Mushroom body defect (Mud; NuMA in vertebrates, Lin-5 in *C. elegans*)[18,20,21]. The Pins complex regulates spindle orientation during mitosis and biased centrosome asymmetry in interphase, manifested in the establishment and maintenance of an apical interphase microtubule organizing center (MTOC). The active interphase MTOC retains the daughter-centriole containing centrosome close to the apical cell cortex and pre-establishes spindle orientation in the subsequent mitosis[22–28].

We found that hybrid cells derived from such NB-GMC fusions independently align the endogenous (neuroblast-origin) and ectopic (GMC-origin) chromosomes at the metaphase plate. Chromosome alignment is implemented through ectopic and endogenous spindles, derived from GMC and NB centrosomes, respectively. We also find that NB and GMC chromosomes segregate independently of each other. We propose that NB-GMC hybrid cells utilize asymmetric centrosome activity in interphase

to retain, and nuclear envelopes to physically separate, epigenetically distinct chromatin in vivo. These findings provide mechanistic insight into how metazoan cells could separate chromosomes of different origins.

## Results

**Acute NB-GMC fusions give rise to viable, mitotically active hybrid cells.** To quantitatively describe chromosome dynamics in hybrid cells we developed an acute cell–cell fusion method in intact larval fly brains. We used a 532 nm pulsed laser to induce a small lesion at the NB–GMC interface, causing the GMC chromatin to enter the neuroblast cytoplasm. Neuroblasts can be distinguished from GMCs based on their size, molecular markers and cell cycle length (Supplementary Fig. 1a). Although larval brains contain type I and some type II neuroblast lineages[29–31], we do not distinguish between the two and will refer to the resulting hybrid cells as NB–GMC hybrids. Targeted mitotic neuroblasts often retained the GMC chromosomes, creating a large apical hybrid cell containing one Dpn$^+$ and one Pros$^+$ nucleus. Most NB–GMC hybrid cells normally localized the contractile ring marker non-muscle Myosin to the cleavage furrow and completed cytokinesis (Supplementary Figure 1b–e). Acute cell fusion can also result in the expulsion of the GMC chromatin, forming GMC – GMC hybrids (see Supplementary Fig. 3c in ref. [32]). Here, we exclusively focus on NB–GMC hybrids (hybrid cells, hereafter).

Fly FUCCI[33] analysis revealed that GMCs contacting NBs were either in G1-S (37%), S-phase (41%) or G2 (22%) (assayed from 110 NB lineages and 763 progeny cells; Supplementary Fig. 2a–d) but could adjust its cell cycle to that of the neuroblast. For instance, G1-S phase GMCs fused with a mitotic neuroblast entered mitosis in contrast to neighboring GMCs (Supplementary Fig. 2e–g; Supplementary Movie 1). We conclude that acute fusions between NBs and GMCs, differing in both their molecular composition and cell cycle stage, can give rise to viable NB-GMC hybrid cells. Furthermore, G1-S phase GMCs can enter mitosis when fused with a mitotic neuroblast.

**NB-GMC hybrid cells independently align NB and GMC chromatin at the metaphase plate.** To better characterize the dynamics of neuroblast (endogenous) and GMC (ectopic) chromosomes during mitosis, we induced cell fusion at different cell cycle stages in wild type neuroblasts, expressing the canonical chromosome marker His2A::GFP. We hypothesized that hybrid cells derived from NB-GMC fusions early in the cell cycle could (1) align only the neuroblast chromosomes at the metaphase plate, (2) congress a mix of neuroblast and GMC chromosomes or (3) separate and align the two chromosome pools at the metaphase plate (Fig. 1a). We found that the endogenous and ectopic chromatin was separated and distinguishable when fusions were induced in early mitosis. Both the ectopic and endogenous chromatin aligned at the metaphase plate (Fig. 1b; Supplementary Movie 2&3). NB-GMC fusions could be induced at all cell cycle stages but GMC chromosomes aligned at the metaphase plate more accurately in hybrid cells derived from interphase or early prophase fusions (Fig. 1c; Supplementary Fig. 3a–c). GMC chromatin usually failed to align in metaphase, anaphase or telophase fusions (Supplementary Fig. 3d–f). We conclude that hybrid cells derived from fusions between interphase/early prophase NBs and GMCs accurately align ectopic and endogenous chromatin at the metaphase plate.

We next asked whether GMC chromatin congresses independently of neuroblast chromatin. To this end, we measured the time between nuclear envelope breakdown (NEB; see methods) and chromosome alignment at the metaphase plate for NB and

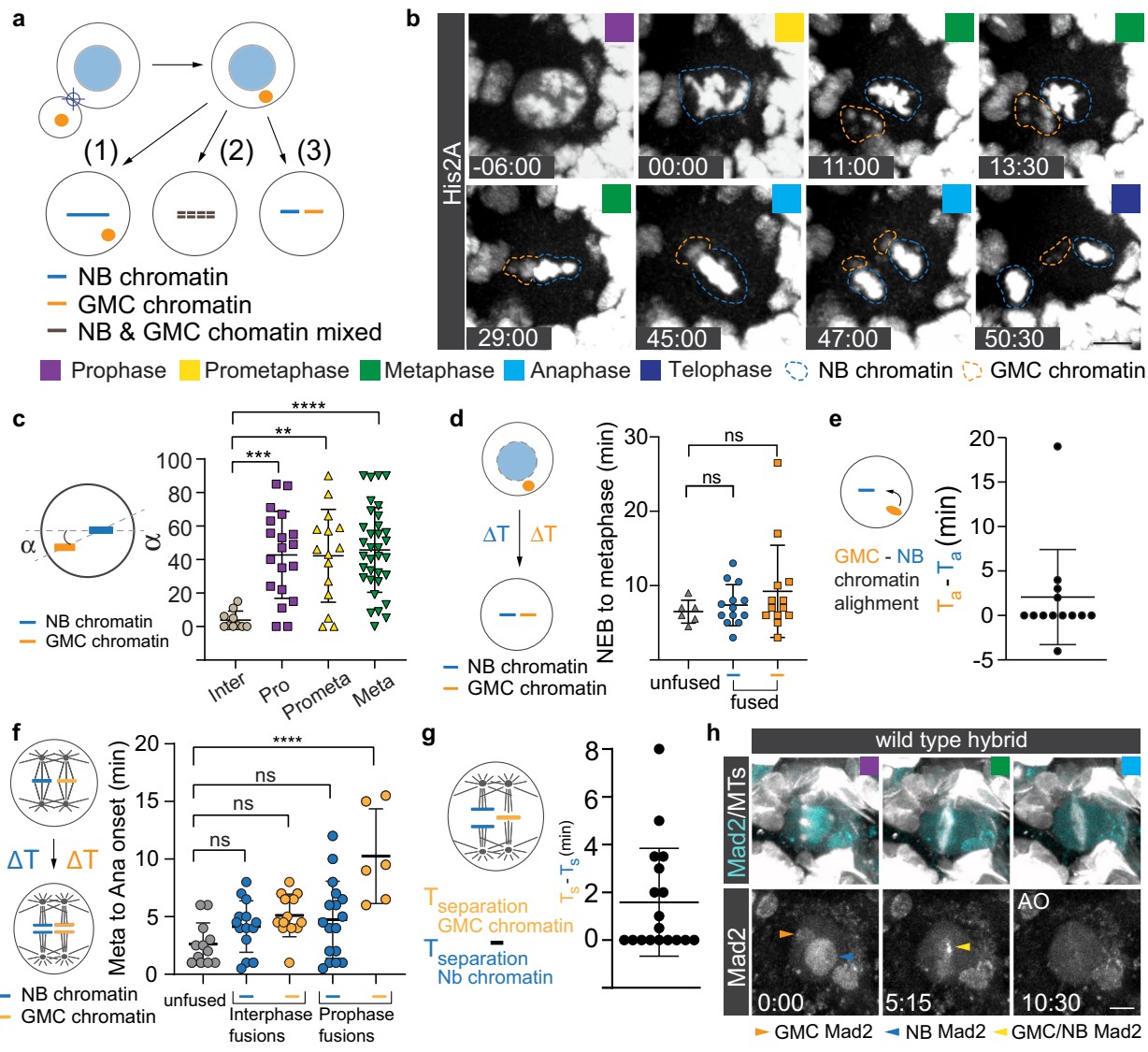

**Fig. 1 NB-GMC hybrid cells can independently align NB and GMC chromatin at the metaphase plate. a** Potential outcomes of NB–GMC fusions: NB–GMC derived hybrid cells could (1) only align neuroblast chromosomes, (2) congress a mix of endogenous and ectopic chromosomes or (3) separately align NB and GMC chromosomes at the metaphase plate. **b** Representative image sequence of a dividing third instar larval NB-GMC hybrid cell obtained from an interphase fusion, expressing the histone marker His2A::GFP (dashed blue circle; endogenous chromatin, dashed orange circle; ectopic chromatin). **c** Alignment of endogenous and ectopic chromatin in hybrid cells derived from interphase (inter), prophase (pro), prometaphase (prometa) or metaphase (meta) fusions were quantified with angle measurements in metaphase. **d** Chromosome alignment time for NB (endogenous; blue circles) and GMC-derived (ectopic) chromosomes (orange squares) compared to unfused control neuroblasts (grey triangles). **e** The time difference ($T_a$; time of alignment) between NB/endogenous (blue line) and GMC/ectopic (yellow ball) chromatin was measured and plotted. **f** Metaphase to anaphase onset was measured for NB/endogenous (blue lines) and GMC/ectopic (orange lines) chromatin, using chromatid separation as a reference. **g** Time difference between NB and GMC chromatin anaphase onset in NB-GMC hybrid cells. **h** Representative images of wild type hybrid cells expressing Mad2::GFP (cyan, top row; white, bottom row) and the spindle marker cherry::Jupiter (white top row). Orange, blue and yellow arrowheads refer to GMC-derived, NB-derived and merged Mad2, respectively. AO; Anaphase onset. Colored boxes represent corresponding cell cycle stages. One-way ANOVA was used for **c**, **d**, **f**. Error bars correspond to standard deviation (SD). *$p < 0.05$, **$p < 0.01$, ***$p < 0.001$, ****$p < 0.0001$. For this and subsequent figures, exact $p$ values and complete statistical information can be found in Supplementary Data 1. Time in min:sec. Scale bar is 5 μm.

ectopic GMC chromatin in hybrid cells derived from interphase and early prophase fusions (Fig. 1d). In most hybrid cells, ectopic and endogenous chromatin was distinguishable either based on differences in location and/or intensity (the reason for these intensity differences is unclear; see Fig. 1b; Supplementary Movie 2 and 3). In untargeted control neuroblasts (unfused), NB chromosomes aligned at the metaphase plate within 6.5 min

after NEB (SD = 1.55; $n = 6$), which is insignificantly faster than the neuroblast chromosomes of hybrid cells (t = 7.4 mins; SD = 2.76; $n = 13$). GMC chromosomes aligned within 9.2 min (SD = 6.2; $n = 13$), statistically not significantly different from unperturbed wild type chromosomes (Fig. 1d). In most NB-GMC hybrids, the endogenous neuroblast and the ectopic GMC chromosomes aligned at the metaphase plate with no significant

time difference. However, in a few cases, ectopic chromatin aligned before or after the neuroblast chromatin, sometimes with a large time difference (Fig. 1b, e). These results suggest that the neuroblast and GMC chromatin can move independently to the metaphase plate in hybrid cells.

**Hybrid cells can independently segregate endogenous and ectopic chromosomes.** Since NB and GMC chromatin can congress independently at the metaphase plate, can the two chromosome pools also segregate independently? We tested this idea by measuring the time between finished chromosome alignment at the metaphase plate and chromosome separation in NB-GMC hybrids expressing His2A::GFP and cherry::Jupiter. Unperturbed control neuroblasts usually initiate anaphase onset within 2.63 min (SD = 1.84; $n = 12$) after chromosomes are aligned at the metaphase plate. In hybrid cells derived from interphase fusions, endogenous and ectopic chromatin entered anaphase 4.14 min (SD = 2.23; $n = 14$) and 5.12 min (SD = 1.85; $n = 13$) after metaphase alignment. Only ectopic chromatin for prophase-induced hybrid cells showed a significantly delayed anaphase onset (Average: 10.25 min; SD = 4.12; $n = 6$) (Fig. 1f). Ectopic chromatin never separated before endogenous chromosomes but entered anaphase with a few minutes' delay (Fig. 1g). These observations suggests that the NB spindle dictates the timing of anaphase onset, presumably via a diffusible 'wait anaphase' signal[8]. To test this idea, we imaged control and hybrid cells expressing Mad2::GFP, a component of the spindle assembly checkpoint (SAC)[8,34]. Control NBs consolidated Mad2::GFP into a bright spot, presumably located at kinetochores, that diminished in intensity shortly before the spindle elongation in anaphase. Hybrid cells either contained a single, or sometimes two Mad2::GFP foci. However, in all observed fusions Mad2::GFP diffused from both foci before the hybrid cells started to elongate the mitotic spindle (Fig. 1h & Supplementary Fig. 4a–d; $n = 7$). We conclude that in NB-GMC hybrids, NB and GMC chromosomes can establish correct MT-kinetochore attachments, thereby fulfilling the spindle assembly checkpoint necessary to enter anaphase. However, given the delays in GMC chromosome separation, we further conclude that ectopic spindles can initiate chromatid separation independently from the endogenous neuroblast spindle, perhaps because the diffusible 'wait anaphase' signal is acting in a distance-dependent manner[8] or because the 'start anaphase' signal overrides the inhibitory signal produced by unattached kinetochores[9].

**Hybrid cells either form parallel double-spindles or an interconnected multipolar spindle.** We next investigated the mechanisms underlying independent NB/GMC chromosome alignment and segregation, considering the following possibilities: ectopic chromosomes could be aligned together with the endogenous chromosomes via a single bipolar spindle, similar to the first zygotic spindle (gonomeric spindle, hereafter) in the *Drosophila* embryo[35]. Alternatively, hybrid cells could form two or more bipolar spindles, which either attach to chromosomes from the neuroblast, and GMC separately or to chromosomes from both cell types (Fig. 2a). Live cell imaging showed that hybrid cells derived from prometaphase, metaphase or anaphase predominantly formed only one mitotic spindle, which attached to the NB's chromatin only (Supplementary Fig. 3c–e; Fig. 2c). Fusions induced in interphase or prophase either formed two clearly separated spindles (hereafter denoted as 'II'; Fig. 2b, d, e; Supplementary Fig. 5a; Supplementary Movies 4 and 5) or interconnected spindles that looked like a 'X' (Fig. 2d, e; Supplementary Fig. 3a; Supplementary Fig. 5b; Supplementary Movie 12). Hybrid cells that had more time between fusion and

NEB were more likely to form 'X'-type spindles (Fig. 2f). 'X'-type spindles usually resolved into a merged spindle by metaphase through positioning of the two MTOCs next to each other at each pole (Supplementary Fig. 5a, b). With some exceptions, II-type spindles often formed at the same time and were initially oriented perpendicular to each other. However, the two separate spindles quickly decreased their inter-spindle distance and angle during metaphase (Fig. 2g–l; Supplementary Movie 5). While spindle architecture offers a potential mechanism for the independent metaphase alignment, and subsequent segregation of NB and GMC chromosomes, we observed that even in neuroblasts with 'X'-type spindles, GMC-derived and NB chromatin remained spatially separated until late metaphase (Supplementary Fig. 5a, b). This suggests that spindle morphology is not the dominant mechanism for the spatial separation of endogenous and ectopic chromatin.

**Ectopic spindles are nucleated from GMC centrosomes.** Mitotic spindles can be nucleated through the centrosome-dependent, chromatin or microtubule pathway but when centrosomes are present, the centrosome pathway prevails[36]. To elucidate the mechanisms underlying ectopic spindle formation, we induced NB-GMC fusions in interphase wild type neuroblasts expressing live centriole (Asterless; Asl::GFP) and spindle (cherry::Jupiter) markers, and assayed centrosome dynamics and spindle formation throughout mitosis. Normal wild type neuroblasts usually contained two Asl::GFP positive centrioles in mitosis, forming a single bipolar spindle. However, in NB-GMC hybrids, we predominantly found four Asl::GFP positive centrioles, two of which were introduced from the GMC (Fig. 3a–c). In all hybrid cells with "II"-type spindles, GMC centrosomes paired with GMC centrosomes and NB centrosomes paired with NB centrosomes. In this case, each cell pole contained a GMC-derived and a NB-derived centrosome, a configuration referred to as 'cis' (Fig. 3d, e and Supplementary Fig. 5a). A small number of 'X'-type spindles either displayed the 'cis' configuration (Fig. 3b, d, f) or a 'trans' configuration, in which each pole contained centrosomes from the same origin cell (Fig. 3d, f & Supplementary Fig. 5b). However, in most cases, we could not attribute the origin of centrosomes when 'X'-type spindles were formed (Fig. 3f). To test whether the centrosome-dependent pathway is used to form double-spindles in hybrid cells, we ablated centrosomes genetically. *asl* mutant neuroblasts lacked functional centrosomes but formed bipolar spindles that poorly converge at the poles (Fig. 3g)[37]. *asl* mutant NB-GMC hybrid cells showed a spindle organization similar to that of unfused *asl* mutant NBs, failing to form clearly separated spindles as shown for wild type hybrid cells. However, GMC-derived chromatin remained separated from NB chromatin until metaphase, when the two pools became indistinguishable (Fig. 3g, h; Supplementary Movie 6 and 7). We conclude that hybrid cells use the centrosome pathway to form parallel (II) or intersecting (X) spindles. Furthermore, these data suggest that parallel spindles form by connecting centrosomes from the same origin cell, whereas intersected spindles are more varied in their centrosome configuration.

**Asymmetric, microtubule-dependent chromatin-centrosome connections tether chromosomes close to the apical neuroblast cortex during interphase.** Considering that hybrid cells with intersected X-type spindles, or unfocused *asl* mutant spindles still separated endogenous from ectopic chromatin, we investigated spindle morphology-independent mechanisms for NB - GMC chromosome separation. During mitosis, microtubules emanate from centrosomes and attach to sister chromatids via kinetochore proteins, localizing to the centromeric region[38]. The centromere-specific H3 variant (Centromere identifier (Cid) in flies)

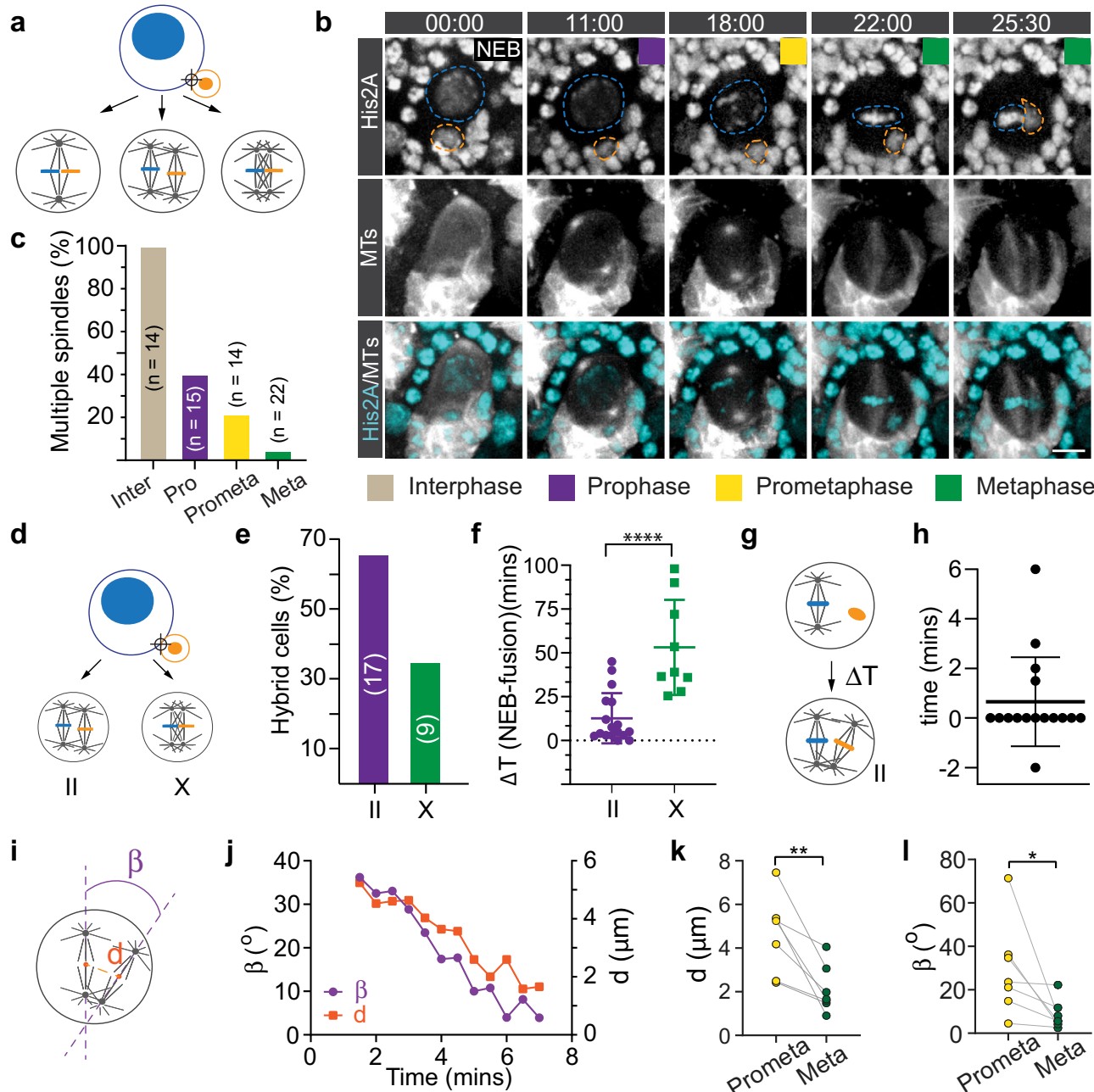

**Fig. 2 Dual spindles align endogenous and ectopic spindles separately at the metaphase plate. a** Hypothetical outcomes of spindle organization after NB-GMC fusions: hybrid cells could align neuroblast and GMC chromosomes either through a single or dual-spindle mechanism. **b** Representative third instar larval NB-GMC hybrid cell, expressing the histone marker His2A::GFP (white in top row; cyan in merged channel below) and the MT marker cherry::Jupiter (white in middle and bottom row). 00:00 refers to the start of nuclear envelope breakdown (NEB). NB- and GMC-derived chromatin is outlined with a blue and orange dashed line, respectively. **c** Quantification of hybrid cells containing dual- or multiple spindles for hybrid cells derived from interphase (inter), prophase (pro), prometaphase (prometa) or metaphase (meta) fusions. **d** Hybrid cells either form clearly distinct parallel (II) or interconnected (X) spindles, quantified in **e**. **f** Time difference between NEB and fusion induction for parallel and interconnected spindles. **g** For hybrid cells with parallel spindles, the time difference between endogenous and ectopic spindle formation was measured and plotted in **h**. **i** For hybrid cells with parallel spindles, spindle angle and inter-spindle distances were measured during mitosis. A representative example is shown in **j**. Quantification of inter-spindle **k** distances and **l** angles at prometaphase and metaphase. Colored boxes represent corresponding cell cycle stages. Error bars correspond to SDs. Unpaired t-test was used in Fig. 2f, n. Two-sided paired t-test was used in Fig. 2k, l. *$p < 0.05$, **$p < 0.01$, ****$p < 0.0001$. Time in min:sec. Scale bar is 5 µm.

colocalizes with centromeres[39]. Since sister chromatids in *Drosophila* male germline stem cells contain asymmetric levels of Cid[11,40], we investigated whether hybrid cell spindles differentiate between endogenous and ectopic chromosomes based on differing levels of Cid. We induced NB-GMC fusions of wild type cells expressing EGFP::Cid[41] in interphase or early prophase and

measured Cid intensity on both GMC and NB chromatin. These measurements did not reveal a significant intensity difference between NB and GMC Cid (Supplementary Fig. 6a). However, we noticed that endogenous EGFP::Cid was localized in very close proximity to the apical centrosome in unperturbed interphase and prophase wild type neuroblasts (Fig. 4a & Supplementary

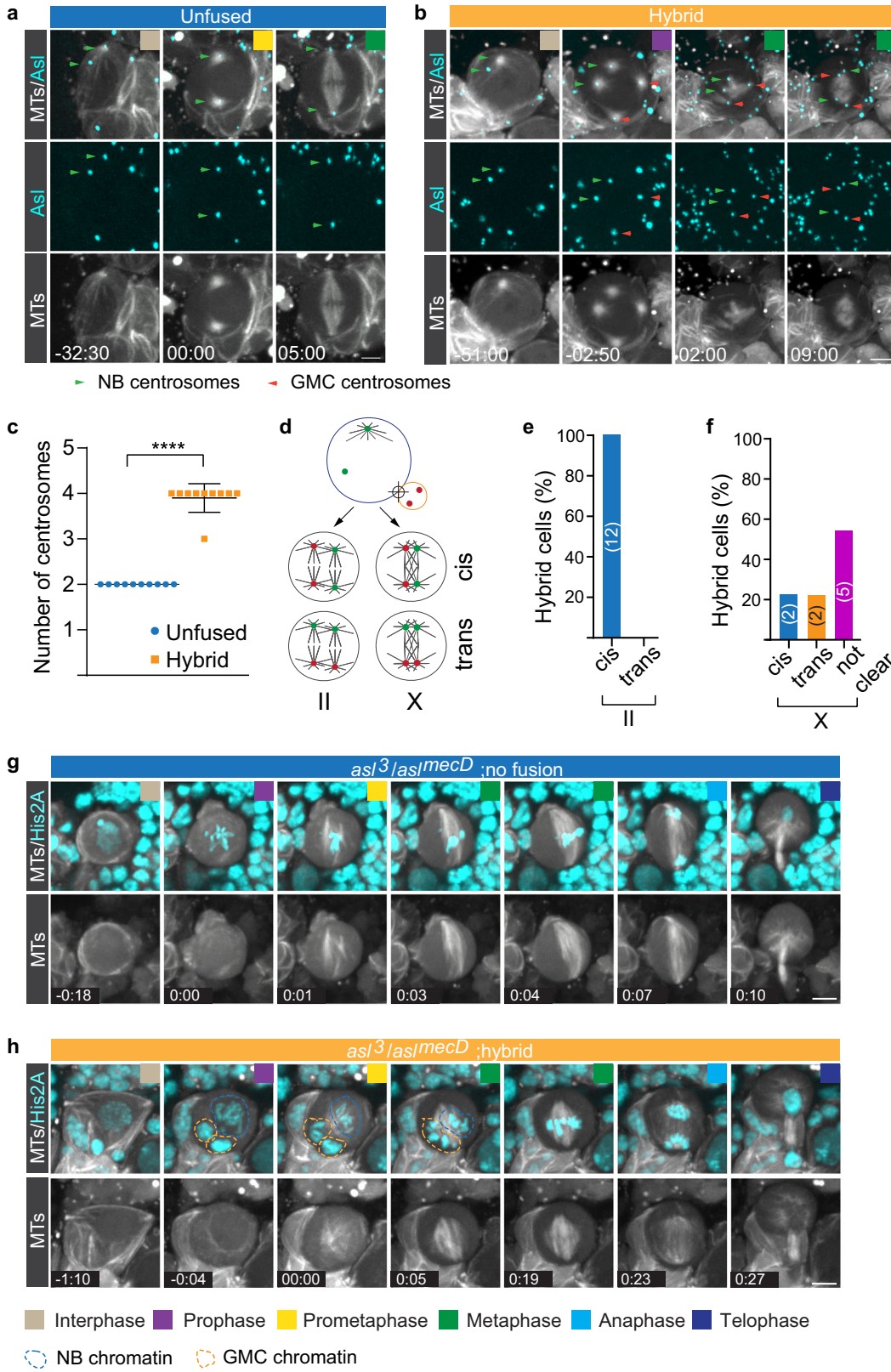

► NB centrosomes   ◄ GMC centrosomes

Interphase   Prophase   Prometaphase   Metaphase   Anaphase   Telophase
NB chromatin   GMC chromatin

Movie 8). EGFP::Cid remained associated with chromatin throughout the neuroblast cell cycle, excluding the possibility that early Cid clusters are not connected with chromatin (Supplementary Fig. 6b & Supplementary Movie 9).

Interphase wild type neuroblasts contain only one active apical microtubule organizing center (MTOC), which anchors the daughter centriole-containing centrosome close to the apical neuroblast cortex. The centrosome containing the mother centriole is inactive in interphase but matures from prophase onward, positioning itself on the basal cell cortex[22,24–28]. We measured the distance of individual EGFP::Cid clusters to the apical and basal centrosome in unperturbed wild type

**Fig. 3 asl mutant hybrid cells contain unfocused spindles. a** Representative third instar larval control NB and **b** NB-GMC hybrid cell, expressing the centriole marker Asl::GFP (cyan) and the spindle marker cherry::Jupiter (MTOCs; white in top and bottom row). Neuroblast-derived and GMC-derived MTOCs were highlighted with green and red arrowheads, respectively. **c** Comparison of centrosome number between unfused wild type and hybrid cells. **d** The two NB-derived centrosomes (green) can either form a bipolar spindle (cis), or pair with ectopic, GMC-derived (red) centrosomes to form a bipolar spindle (trans). **e**, **f** Quantification of cis and trans spindles in wild type hybrid cells. **g** Unfused *asl* mutant neuroblasts and **h** *asl* mutant hybrid cells, expressing the chromatin marker His2A::GFP (cyan) together with the spindle marker cherry::Jupiter (white). Colored boxes represent corresponding cell cycle stages. Error bars correspond to SDs. Unpaired t-test was used in Fig. 3c. ****p < 0.0001. Time in mins:secs. Scale bar is 5 mm.

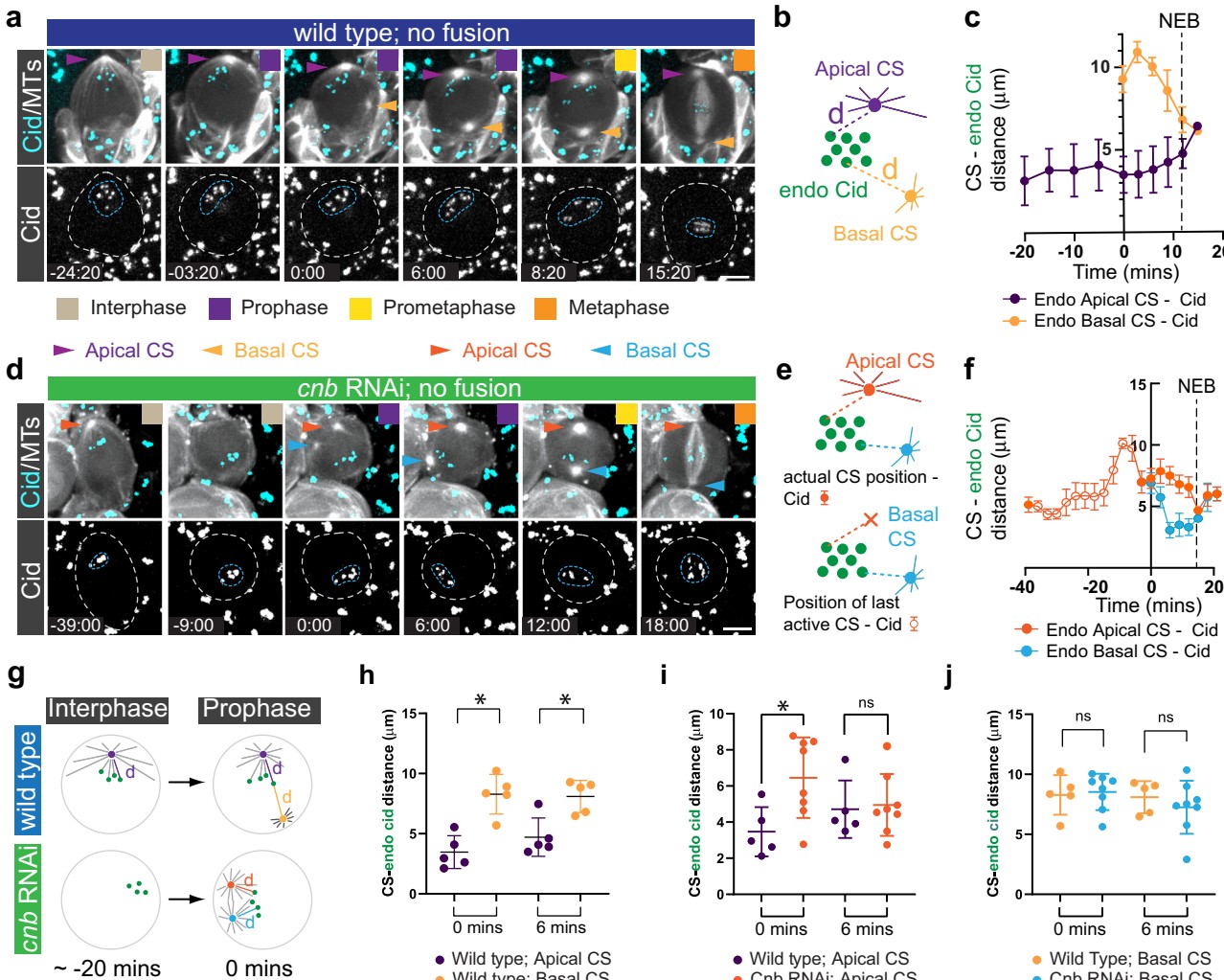

**Fig. 4 Biased MTOC activity retains Cid in the apical neuroblast hemisphere during interphase and early mitosis. a** Representative third instar larval neuroblast expressing the centromere specific Histone-3 variant marker, EGFP::Cid (cyan; top, white; bottom) and the microtubule marker cherry::Jupiter (white; top). Colored boxes represent corresponding cell cycle stages. **b** The distance (purple and yellow dashed lines) between the apical (purple) and basal (yellow) centrosome (CS), and individual Cid clusters (green circles) were measured throughout the cell cycle and plotted in **c**. **d** Representative third instar larval neuroblast expressing *cnb* RNAi, cherry::Jupiter (white; top) and EGFP::Cid (cyan; top, white; bottom row). Orange arrowheads highlight the apical MTOC. Blue arrowheads highlight the maturing basal MTOC. Note that *cnb* mutant neuroblasts lose the active MTOC in interphase (-9:00). The blue and white dashed circle highlights Cid clusters and the cell outline, respectively. 'Apical' centrosome refers to the centrosome destined to be positioned on the apical cortex, whereas 'basal' centrosome will be inherited by the basal GMC. **e** CS – Cid distance measurements were performed in *cnb* RNAi expressing NBs. Once the apical CS disappeared in interphase, the last detectable position was used as a reference point (orange cross in the schematic below; open circles in the graph). **f** CS – Cid measurements for *cnb* RNAi expressing NBs. Closed arrows refer to actual CS – Cid measurements. Open circles denote Cid – previous active CS measurements. **g** Wild type neuroblasts maintain apical CS – Cid attachments in prophase, due to asymmetric MTOC activity and microtubule-dependent interphase centrosome – Cid attachments. *cnb* RNAi expressing neuroblasts lose MTOC activity in interphase, randomizing the position of Cid clusters. When centrosomes mature again in prophase, both centrosomes simultaneously attach to Cid clusters. **h** Centrosome – Cid distance of an unperturbed wild type neuroblast at the time of basal centrosome maturation (0 min) and 6 min thereafter. **i**, **j** Cid – centrosome distance measurements, comparing wild type with *cnb* RNAi expressing neuroblasts. Error bars correspond to SDs. Two-sided paired or unpaired t-test was used in Fig. 4h, i, j. ns; no significance. *p < 0.05. Time in mins:secs. Scale bar is 5 µm.

neuroblasts and found that Cid was always in close proximity to the apical centrosome during interphase. After nuclear envelope breakdown (NEB), Cid moved progressively towards the metaphase plate. Once the basal centrosome appeared (0 mins), Cid was still closer to the apical than the basal centrosome and this distance asymmetry was also observed 6 min after the appearance of the basal centrosome. After nuclear envelope breakdown (NEB), Cid moved progressively towards the metaphase plate (Fig. 4a, b, c, h).

The proximity of EGFP::Cid clusters to the active interphase MTOC suggests a microtubule-dependent chromatin localization mechanism. Indeed, wild type neuroblasts treated with the microtubule-depolymerizing drug colcemid showed a strong correlation between apical MTOC activity and Cid localization: as MTs depolymerized after colcemid addition, Cid progressively moved away from the apical cortex towards the cell center (Supplementary Figure 6c, d & Supplementary Movie 10). To test whether polarized Cid localization specifically depends on interphase MTOC activity, we removed the centriolar protein Centrobin (Cnb; CNTROB in humans). Neuroblasts lacking Cnb fail to maintain an active apical interphase MTOC but regain normal MTOC activity during mitosis[26]. Neuroblasts expressing cnb RNAi lost apical Cid localization after the apical centrosome downregulated its MTOC activity. However, maturing centrosomes reappeared in close proximity to Cid in prophase (Fig. 4d–g & Supplementary Movie 11). Cid's proximity to the apical MTOC ('apical' refers to the centrosome destined to move to the apical cortex) in cnb RNAi expressing neuroblasts was much more varied compared to wild type. At 6 min after centrosome maturation onset, Cid – apical MTOC distance was comparable to wild type, as were Cid – basal centrosome distance relationships (Fig. 4i, j). We conclude that Cid's polarized localization depends on interphase MTOC activity.

**Asymmetric centrosome-chromatin connections contribute to the separation of endogenous and ectopic chromatin in hybrid cells in early mitosis.** Based on these observations, we hypothesized that the physical separation between endogenous NB and ectopic GMC chromosomes in NB-GMC hybrid cells could be dependent on the active apical MTOC. To test this hypothesis, we tracked centrosome and Cid movements in wild type hybrid cells and analyzed Cid localization in relation to the endogenous and ectopic centrosomes. Similar to unperturbed wild type neuroblasts, endogenous Cid was also localized in close proximity to the endogenous apical MTOC in wild type hybrid cells (Fig. 5a, c, & Supplementary Movie 12; 'apical' refers to the centrosome destined to segregate into the large apical sibling cell). Ectopic Cid, however, appeared closer to ectopic centrosomes (Fig. 5a, d & Supplementary Movie 12; '0' refers to the appearance of the ectopic centrosomes).

We next attempted to randomize the distance of endogenous and ectopic Cid relative to centrosomes by inducing fusions in cnb RNAi expressing neuroblasts, since loss of interphase MTOC activity released endogenous Cid from the apical centrosome (Fig. 4d–j and Supplementary Fig. 6c, d). In contrast to wild type hybrid cells, endogenous Cid is roughly equidistant to the endogenous NB and ectopic GMC MTOCs in cnb RNAi expressing hybrid cells at '0' min (Fig. 5b, e & Supplementary Movie 13). However, ectopic Cid was still closer to the ectopic centrosomes than to the endogenous apical centrosome in cnb RNAi expressing hybrid cells (Fig. 5b, f & Supplementary Movie 13). In both wild type and cnb RNAi expressing hybrid cells, ectopic GMC and basal NB MTOCs were about equidistant to endogenous Cid, but ectopic centrosomes were closer to ectopic Cid (Supplementary Fig. 7a–e).

We next asked whether biased MTOC activity influences the physical separation between endogenous and ectopic chromosomes and tracked endogenous and ectopic EGFP::Cid after induced cell fusion. To this end, we measured the distance between endogenous and ectopic Cid in hybrid cells, using endogenous Cid–Cid distance as a baseline and determined when endo-ecto CID distance was indistinguishable from this endo-endo CID distance (defined as 'T'). We then calculated and plotted the time difference between T and NEB (defined as $\Delta T$; Fig. 5g; see methods). In most wild type hybrids, endogenous and ectopic Cid remained physically well separated until ~10 min after NEB (mean $\Delta T = 9.6$ min; SD = 5.7 mins; $n = 9$). In cnb RNAi expressing hybrid cells, the endogenous Cid-Cid and ectopic Cid – endogenous Cid distances were similar already ~1 min after NEB (mean = 0.73 min; SD = 7.9; $n = 15$), which is significantly earlier compared to wild type. Hybrid cells derived of fusions between cnb PACT expressing neuroblasts, which contain only active interphase MTOCs[26], and GMCs showed an intermediate phenotype (Fig. 5h).

Altering MTOC activity bias also influenced spindle architecture in hybrid cells. In contrast to wild type hybrid cells, cnb RNAi or cnb PACT hybrid cells predominantly formed 'X'-type spindles with no statistically significant time difference between fusion induction and NEB (Figs. 2e, f and 5i, j). We conclude that the apical, endogenous MTOC retains endogenous Cid close to the apical cortex in wild type hybrid cells, which affects the timing of the physical separation between NB and GMC chromatin in hybrids.

**Nuclear envelopes provide a barrier between endogenous and ectopic chromatin in hybrid cells.** We next hypothesized that the separation of endogenous and ectopic chromatin could be supported by nuclear envelopes. Neuroblasts undergo semi-closed mitosis, mostly retaining a matrix composed of nuclear envelope proteins around the mitotic spindle[42,43]. We imaged wild type neuroblasts with the nuclear envelope marker Lamin (UAS-Lamin::GFP) and confirmed that unperturbed wild type neuroblasts contain a nuclear envelope matrix surrounding the mitotic spindle during mitosis (Fig. 6a). Similarly, wild type hybrid cells contain two nuclear envelopes during mitosis, which remain separate until early anaphase. However, the separation between the NB and introduced GMC NE became diffuse during anaphase (Fig. 6b; Supplementary Movie 14).

The observed asymmetric interphase chromatin localization could be mediated via direct or indirect microtubule—Cid attachments. In the case of a direct attachment, we would expect to see microtubules penetrating the nuclear envelope prior to mitosis entry. To address this question, we stained wild type neuroblasts expressing Lamin::GFP with anti-Tubulin and anti-CID. However, we only saw microtubules entering the nuclear envelope in prometaphase neuroblasts, but not before, indicating that the MT-Cid connections observed during interphase are most likely indirect. (Fig. 6c). Different imaging techniques will be needed to exclude the possibility that we simply do not see MTs penetrating the NE before prometaphase.

These data suggest that nuclear envelopes establish a physical barrier between NB and GMC chromatin at least until metaphase in hybrid cells.

**Hybrid cells show a variety of chromosome missegregation defects.** Finally, we assessed the accuracy of chromosome segregation in wild type hybrid cells. Using the canonical chromosome marker His2A::GFP we detected chromosome missegregation—ranging from lagging chromosomes to chromosome bridges—in wild type hybrid cells (Fig. 7a, b, & Supplementary Movie 15).

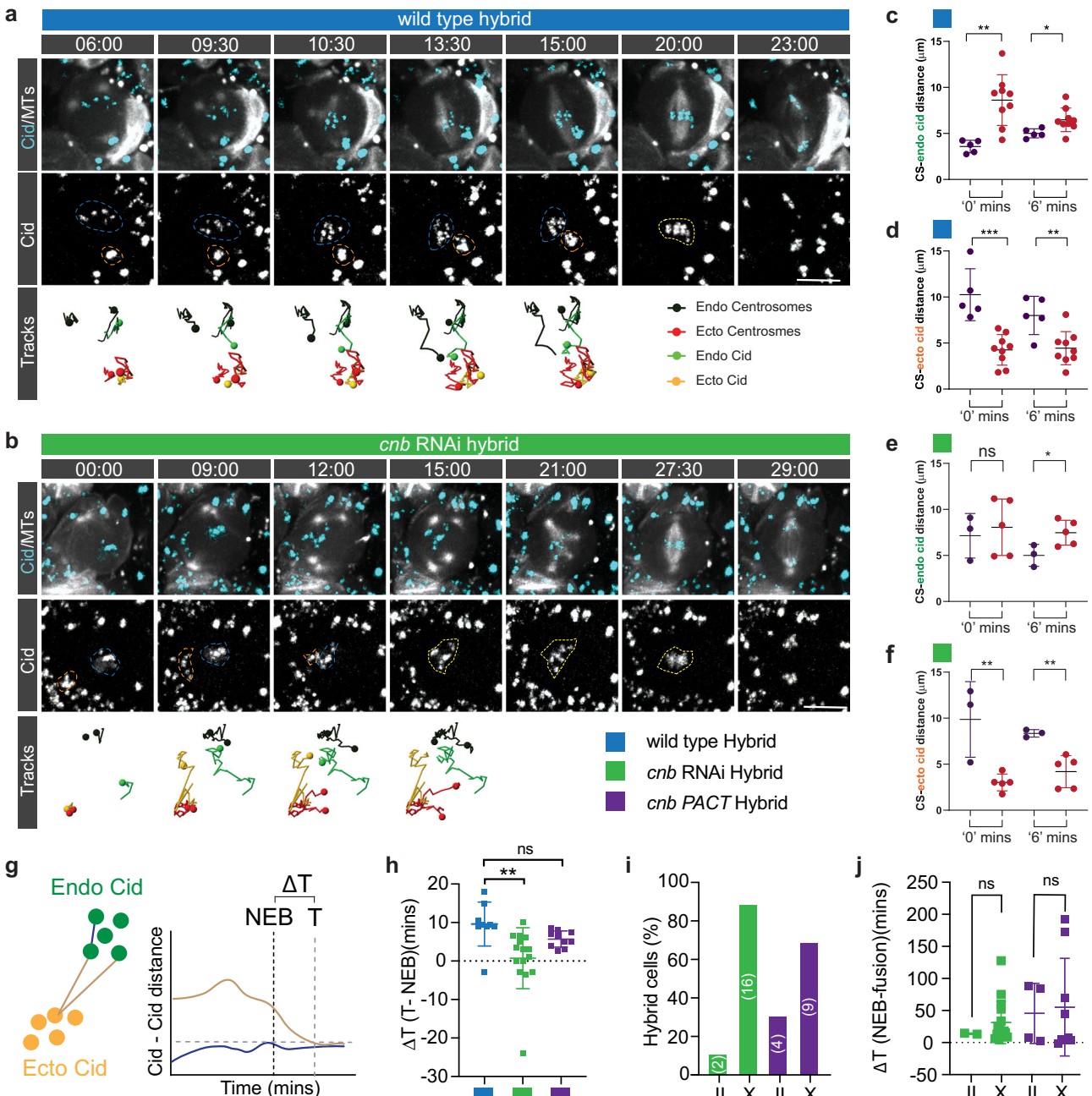

**Fig. 5 Asymmetric microtubule dependent centrosome-chromatin attachments contribute to the separation of endogenous and ectopic chromosomes in hybrid cells.** Representative third instar larval **(a)** wild type or **(b)** *cnb* RNAi hybrid cell, expressing EGFP::Cid (top row; cyan, bottom row; white) and the microtubule marker cherry::Jupiter (white; top row). Neuroblast-derived and GMC-derived Cid clusters are outlined with a blue and orange dashed line, respectively. Indistinguishable GMC and NB Cid clusters are highlighted with yellow dashed circles. Centrosome and Cid tracks are shown below the snapshots. Scatter plot showing the distance of endogenous Cid in relation to GMC- and NB-derived centrosomes (CS) for **c** wild type (blue square) and **e** *cnb* RNAi expressing hybrid cells (green square). Scatter plot showing the distance of ectopic Cid in relation to GMC- and NB-derived centrosomes (CS) for **d** wild type (blue square) and **f** *cnb* RNAi expressing hybrid cells (green square). **g** The distance of endogenous-to-ectopic Cid, or endogenous-to-endogenous Cid was measured and Delta T – the time difference between NEB and when Ecto-Endo Cid distance is the same as Endo-Endo Cid distance (T) - plotted for **h** wild type, *cnb* RNAi and *cnb* PACT. **i** Percentage of II vs X spindles in *cnb* RNAi and *cnb* PACT. **j** Scatter plot showing the time between fusion induction and NEB for *cnb* RNAi or *cnb* PACT hybrid cells containing 'II' and 'X' spindles. Error bars correspond to SDs. **h, j** Two-sided unpaired t-test. ns; no significance. *$p < 0.05$, **$p < 0.01$, ***$p < 0.001$, ****$p < 0.0001$. Time in mins:secs. Scale bar is 10 µm.

Chromosome segregation defects can result in aneuploidy and micronuclei formation[44]. In most cases, hybrid cells fused both nuclei into one, forming synkaryons, while some cells formed heterokaryons (hybrid cells containing two nuclei of different origins), and a small percentage of hybrid cells formed

micronuclei[15]. Synkaryon formation is most likely not a consequence of failed cytokinesis but due the merging of both nuclei (Fig. 7c, d; Supplementary Fig. 8).

We conclude that NB-GMC hybrids missegregate chromosomes, leading to the formation of micronuclei, heterokaryons or

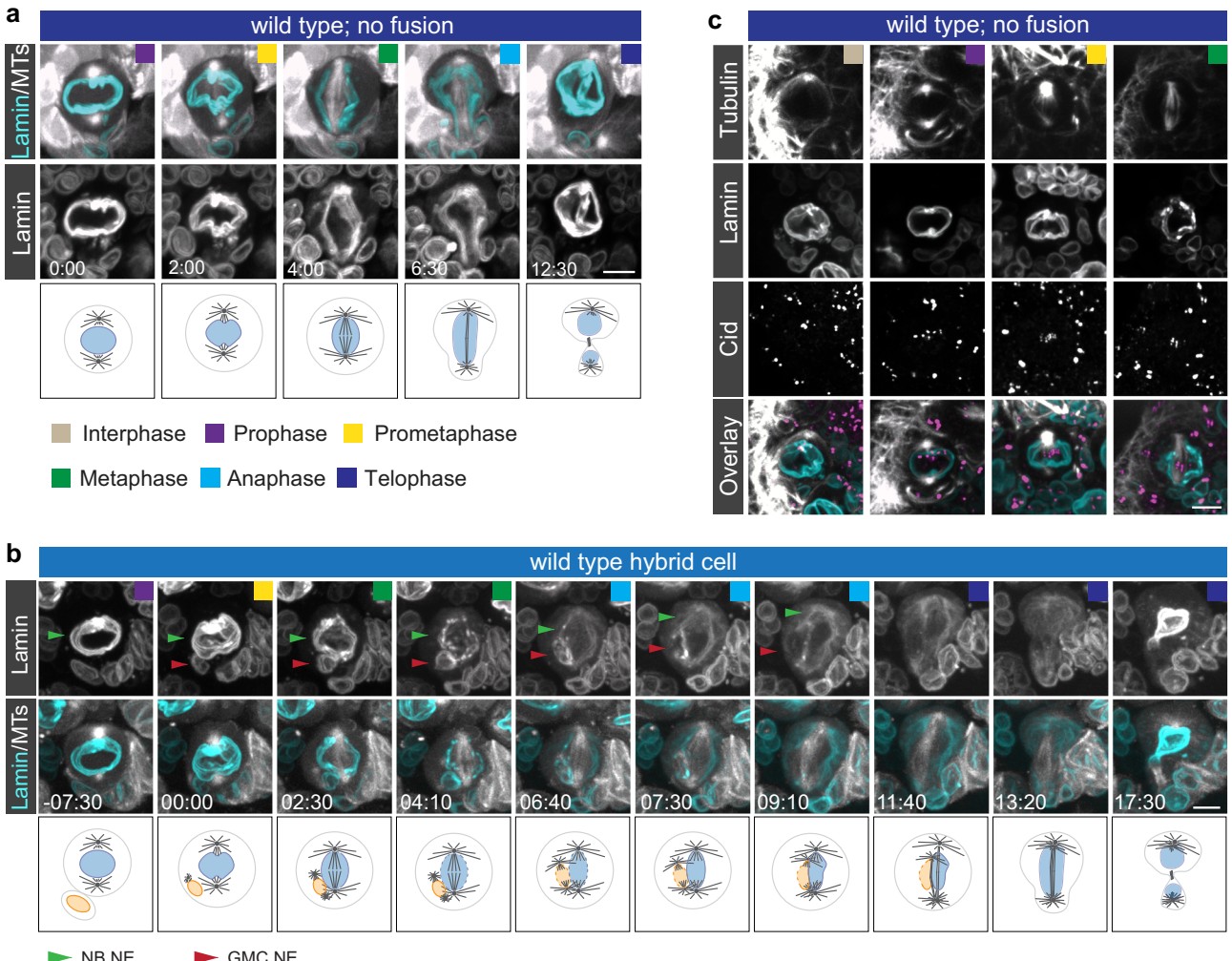

**Fig. 6 Nuclear envelopes contribute to the separation of endogenous and ectopic chromatin in NB – GMC hybrid cells. a** Unfused wild type neuroblasts or **b** hybrid cell expressing the nuclear envelope marker Lamin::GFP (cyan on top; white in the bottom row) and the spindle marker cherry::Jupiter (white on top). Schematics are shown below the image sequence. Green arrow; nuclear envelope (NE) of the neuroblast (NB). Red arrow; nuclear envelope (NE) of the ganglion mother cell (GMC). **c** Wild type neuroblasts expressing Lamin::GFP (cyan in merge) and stained for anti-Tubulin (white in merge) and anti-Cid (magenta in merge). Time in mins:secs. Scale bar is 5 μm.

synkaryons. We find no evidence that heterokaryons or synkaryons are a consequence of failed cytokinesis and appear to be a result of nuclear fusion.

## Discussion

Cell–cell fusion can occur under normal physiological conditions and has also been implicated in malignancy[16]. How hybrid cells recognize and separate endogenous and ectopic chromosomes during mitosis is not known. Here, we acutely induce cell-cell fusions in vivo between neural stem cells and differentiating GMCs (or less frequently INPs) in the developing larval fly brain. Previously performed cell fusion experiments with cultured cells of the same type have revealed important conceptual and molecular information about the mechanisms of the spindle assembly checkpoint[8,9]. NB-GMC fusions do not naturally occur and laser-based acute fusions could induce some unintended damage. However, because we fuse molecularly distinct cell types, containing epigenetically different chromatin, our system is useful to reveal basic mechanisms of chromosome recognition and separation. We found that NB-GMC derived hybrid cells keep endogenous neuroblast chromosomes separated from the introduced ectopic GMC chromosomes and align them independently

at the metaphase plate. Hybrid cells usually contain two neuroblast- and two GMC-derived centrosomes, which either form two distinct parallel, or interconnected mitotic spindles. Spindle architecture depends on the timing of fusion induction because fusions performed long before NEB predominantly created interconnected, X-like spindles, whereas fusions induced shortly before NEB generated clearly distinct, II-like looking spindles. Regardless of spindle morphology, centrosome migration and spindle realignment position both spindles next to each other during metaphase, thereby congressing the neuroblast- and GMC-derived chromosomes at the metaphase plate. Hybrid cells subsequently enter anaphase although NB and GMC chromosomes often do not migrate to the cell poles at the same time. The distinct anaphase onset between NB and GMC chromosomes could be due to the previously implicated diffusible 'wait anaphase' signal that acts in a distance-dependent manner[8] or because the 'start anaphase' signal overrides the inhibitory signal produced by unattached kinetochores[9]. However, the fact that NB and GMC chromosomes can initiate segregation autonomously suggests that hybrid cell spindles can operate independently, implying that NB and GMC chromatin remain separated throughout anaphase. This separation could be achieved through

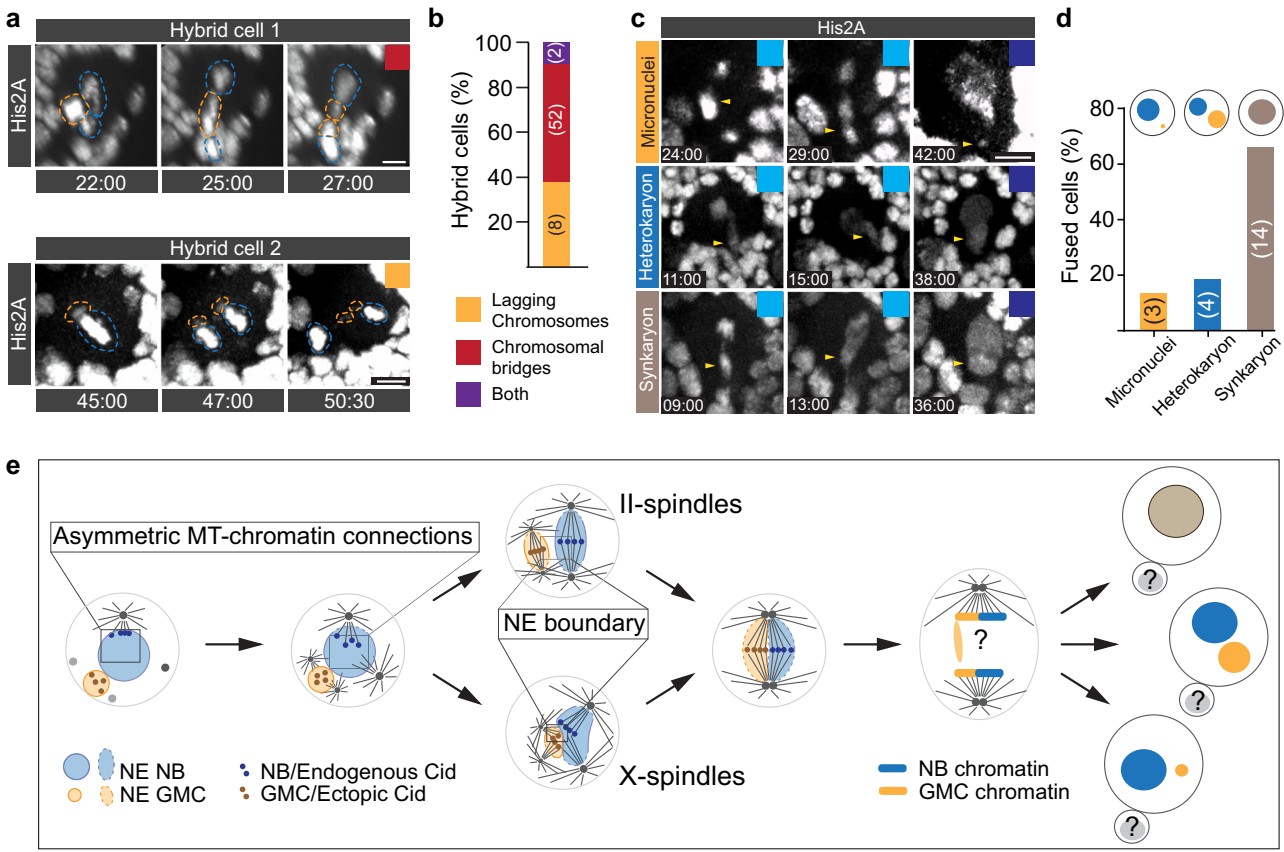

**Fig. 7 Ectopic chromosomes in hybrid cells segregate erroneously. a** Representative examples of delayed (top row) and simultaneous (bottom row) segregation of endogenous (blue dashed circle) and ectopic (orange dashed circle) chromosomes in wild type hybrid cells expressing the chromatin marker His2A::GFP. **b** Bar graph showing percentage of NB-GMC hybrid cells with lagging chromosomes, chromosomal bridges, or both. **c** Representative third instar larval NB-GMC hybrids expressing histone marker His2A::GFP showing missegregating chromatids (yellow arrowheads) during anaphase, resulting in micronuclei (top row), heterokaryon (middle row) or synkaryon (bottom row) formation. Time stamps are in relation to NEB (=0:00). **d** Bar graph quantifying the percentage of fused cells with micronuclei, heterokaryons or synkaryons. **e** Summary and model. Time in mins:secs. Scale bar is 5 μm.

nuclear envelopes. It has previously been described that fly neuroblasts undergo semi-closed mitosis[42,43] and we found that NB-GMC hybrid cells contain independent nuclear envelopes, differing in size, which appeared to merge during late metaphase or early anaphase. Because the NE signal fades during anaphase, it is difficult to see whether physical barriers between the NB and GMC-derived spindle persist.

We also observed that the centromere-specific H3 variant is connected—either directly or indirectly—to the active interphase MTOC, located close to the apical NB cortex. Apical MTOC-Cid connections ensure that NB Cid remains located in close proximity to the nuclear envelope facing the apical MTOC. This connection keeps chromatin from floating around the nucleus and prevents it from getting in close proximity to the GMC chromatin. Because GMCs are predominantly clustered on the basal side of the neuroblast, the observed NB-GMC chromatin separation could be an artefact of this cell arrangement. However, conditions that removed biased MTOC activity caused NB and GMC Cid clusters to lose their physical separation prematurely, although nuclear envelopes still prevented a complete mixing of the two chromatin pools. Thus, it is unlikely that NB-GMC arrangements, and the subsequent location of the GMC chromatin after cell-cell fusion, is the dominant mechanism for the observed physical chromatin separation.

Finally, we showed that NB-GMC hybrid cells display chromosome segregation errors in anaphase, which could be due to incompletely replicated GMC chromosomes, spindle morphology defects or erroneous MT-kinetochore attachments. However, NB-GMC hybrid cells successfully complete cytokinesis and predominantly form synkaryons as well as heterokaryons or micronuclei with less frequency.

Based on these observations, we propose that the separation, independent alignment and segregation of NB and GMC chromosomes depends on (1) biased interphase centrosome activity, connecting the active apical MTOC with NB-chromatin throughout interphase and early mitosis, and (2) nuclear envelopes, imposing physical boundaries between the NB and GMC chromatin (Fig. 7e). This model builds on previous observations, showing that wild type interphase neuroblasts contain one active MTOC that remains stably anchored to the apical cell cortex[24,28]. In contrast to yeast, where chromosomes make dynamic attachments to microtubules in G1[45], this is generally not the case in other metazoan cells[2]. Chromatin can be connected with centrosomes through the LINC complex[46], potentially implicating the SUN domain protein Klaroid[47] and the KASH-domain protein Klarsicht[46,48] in asymmetric chromatin clustering.

The functional significance of these interphase microtubule-chromatin connections in normal neuroblasts are not known but could be similar to *Drosophila* male germline stem cells (GSCs), where a single active centrosome connects to chromosomes in prophase, a potential mechanism for biased chromatid segregation[11,49]. Centromeres have also been found to be confined to specific nuclear locations in other organisms[50,51], suggesting other important cellular functions. The biased centromere

localization we report here could reflect a previously observed pattern of chromosome organization referred to as the Rabl conformation, although we do not report on neuroblast telomere positioning here[51–54]. In nuclei exhibiting a Rabl conformation, the polarized arrangement of chromosomes established in telophase, with centromeres clustered on one side of the nucleus and telomeres clustered on the opposite side, is retained in interphase cells. In *Drosophila*, the Rabl configuration has been reported in the nuclei of embryonic and salivary gland cells and has been suggested in CNS cells from third instar larvae[55–60]. Asymmetric centromere positioning has been reported in cultures S2 cells, but centromeres are more dispersed than the localization we report here in neuroblasts and telomeres do not clearly cluster[61].

Although GMC and NB chromatin are epigenetically distinct[43] we currently have no evidence to suggest that hybrid cells actively distinguish between the two chromatin pools. Chromosome separation is also observed during the first zygotic division after fertilization in both invertebrates and vertebrates and it is equally unclear whether the underlying mechanisms entail active separation processes. For instance, during gonomeric-type fertilization the two pronuclei do not fuse and stay side by side, each independently forming a mitotic apparatus with a haploid chromosome group[7]. Zygotic divisions with gonomeric spindles occur in different species such as the firebug *Pyrrhocoris apterus*, *Drosophila melanogaster*, the silkmoth *Bombyx mori*, the cricket *Gryllus bimaculatus*, or the copepod *Cyclops*[7,35,62,63]. Similarly, dual-spindles separate the paternal from the maternal chromatin during the first zygotic division in mice and bovines but in subsequent single-spindle divisions, genome compartmentalization is lost[6,64].

The functional significance of chromosome separation in either somatic hybrid cells or during the first zygotic division are unknown. However, it is noteworthy that the ability to form a spindle from a female pronucleus and to continue subsequent mitoses makes haploid parthenogenesis possible for social insects[7]. To investigate the consequences of altered chromatin compartmentalization in either hybrid cells or zygotes, acute microtubule and nuclear envelope manipulations will have to be developed to mix epigenetically distinct chromatin with spatiotemporal resolution. Future studies will further aim at elucidating the mechanisms and significance of asymmetric chromatin localization.

## Methods

**Fly strains**. Mutant alleles, transgenes and fluorescent markers: *worGal4, UAS-mCherry::Jupiter*[65]; *worGal4, UAS-mCherry::Jupiter, Sqh::GFP*[66]; *His2A::GFP* (Bloomington stock center); *UAS-lamin::GFP* (Bloomington stock center); *UAS-mCherry::CAAX, UAS-iLID::CAAX:;mCherry* (this work); *EGFP::Cid*[41]; *pUbq-Asl::GFP*[67]; *worgal4, UAS-mCherry::Jupiter, Asl::GFP* (this work); *pros::EGFP* (endogenously tagged with CRISPR; this work); *Mad2::GFP*[68]; *cnb*[GD11735] RNAi line (v28651)[69]; UAS-GFP.E2f1.1–230, UAS-mRFP1.NLS.CycB.1–266}17/TM6B, Tb[33] (Bloomington stock center); *pUASp-YFP::PACT-Cnb*[26], *asl*[2,70], *asl*[MecD][67].

**Antibodies**. The following primary antibodies were used for this study: mouse anti-α-Tub (DM1A, Sigma; 1:2500; Cat# T6199), rabbit anti-Cid (1:500; Active-motif; Cat#39719). Secondary antibodies were from Invitrogen.

**Immunohistochemistry**. Third instar larval brains were dissected in Schneider's medium for no longer than 20 min. Brains were fixed in 4% formaldehyde in PEM (100 mM PIPES pH 6.9, 1 mM EGTA and 1 mM $MgSO_4$). After fixing, the brains were washed and blocked using 1× PBSBT. Subsequently, brains were incubated overnight at 4 °C in the primary antibody dilution in 1× PBSBT (1× PBS, 0.5% Triton-X 100, 1% BSA). Brains were then washed three times with 1× PBSBT and incubated overnight in the secondary antibody dilution prepared in 1X PBSBT. Brains were again washed three times in 1× PBT (1× PBS, 0.5% Triton-X 100) before mounting in vectashield.

**Generation of pros::EGFP with CRISPR**. Target specific sequences with high efficiency were chosen using the CRISPR Optimal Target Finder (http://tools.flycrispr.molbio.wisc.edu/targetFinder/), the DRSC CRISPR finder (http://www.flyrnai.org/crispr/), and the Efficiency Predictor (http://www.flyrnai.org/evaluateCrispr/) web tools. Sense and antisense primers for these chosen sites were then cloned into pU6-BbsI-ChiRNA[71] between BbsI sites. To generate the replacement donor template, EGFP and 1 kb homology arms flanking the insertion site were cloned into pHD-DsRed-attP (Addgene plasmid #51019) using Infusion technology (Takara/Clontech). Injections were performed in house. Successful events were detected by DsRed-positive screening in the F1 generation. Constitutively active Cre (BDSC#851) was then crossed in to remove the DsRed marker. Positive events were then balanced, genotyped, and sequenced.

**Live cell imaging and acute cell-cell fusion**. Imaging medium consists of Schneider's insect medium (Sigma-Aldrich S0146) mixed with 10% BGS (HyClone). Third instar larvae were dissected in imaging medium and the brains were transferred into a μ-slide Angiogenesis or μ-slide 8 well (Ibidi). Live samples were imaged with an Intelligent Imaging Innovations (3i) spinning disc confocal system, consisting of a Yokogawa CSU-W1 spinning disc unit and two Prime 95B Scientific CMOS cameras. A 60x/1.4NA oil immersion objective mounted on a Nikon Eclipse Ti microscope was used for imaging. Live imaging voxels are 0.22 × 0.22 × 0.75-1 μm (60x/1.4NA spinning disc).

Neuroblast-GMC fusions were induced using a 3i Ablate! ablation system, consisting of a 532 nm pulsed laser. We used a pulse width of 7 ns, targeting the membrane interface between the neuroblast and the adjacent GMC.

Nuclear envelope breakdown (NEB) was used as a cell cycle reference point. NEB was determined based on His2A::GFP or Cherry::Jupiter signal. For instance, His2A::GFP can be detected in the entire nucleus prior to NEB but quickly localizes to chromatin as MTs penetrate the NE. Microtubules labelled with Cherry::Jupiter enter the nucleus at the start of NEB. The first frame of MTs entering the nuclear space is thus defined as NEB.

Manual tracking, observed His2A::GFP intensity differences and/or physical spacing allowed to distinguish between endogenous and ectopic chromosomes and/or centrosomes until at least early metaphase. Movies were acquired with time resolutions ranging from 30 s to 4 min (depending on the experiment).

**Colcemid treatment**. Dissected brains were incubated with Colcemid (Sigma) in live imaging medium at a final concentration of of 25 μg mL$^{-1}$.

**Image processing and measurements**. Live cell images were processed using Imaris x64 8.3.1 and image J. For angle and distance measurements, the coordinates for the two spindle poles were determined in Imaris. From these coordinates, angles and distances between spindles were derived based on the calculations outlined below.

*Angle between spindles*: $\theta = \cos^{-1}\frac{n \cdot e}{|n||e|}$

Dot product: $n \cdot e = (X1*X2) + (Y1*Y2) + (Z1*Z2)$

Magnitude of vectors: $|n| = \sqrt{X_1^2 + Y_1^2 + z_1^2}$ $|e| = \sqrt{X_2^2 + Y_2^2 + z_2^2}$

Where **n** corresponds to the spindle vector: **n** (x1, y1, z1) = (N1-N1′, N2-N2′, N3-N3′) and **e** to the ectopic spindle vector: **e** (x2, y2, z2) = (E1-E1′, E2-E2′, E3-E3′).

N1, N2, N3 and N1′, N2′ and N3′ are coordinates of the two poles of the NB spindle. Similarly, E1, E2, and E3 and E1′, E2′ and E3′ are coordinates of the ectopic spindle poles.

*Distance between spindle vectors*. The midpoints of the two spindle vectors are calculated from coordinates of the poles on either side of the respective spindle. This is followed by calculating the distance between these midpoints.

Midpoint of the NB spindle vector = $\left(\frac{N_1+N_1'}{2}, \frac{N_2+N_2'}{2}, \frac{N_3+N_3'}{2}\right) = (M_1, M_2, M_3)$

Midpoint of the GMC spindle vector = $\left(\frac{G_1+G_1'}{2}, \frac{G_2+G_2'}{2}, \frac{G_3+G_3'}{2}\right) = (m_1, m_2, m_3)$

Distance between these two points = $\sqrt{(M_1-m_1)^2 + (M_2-m_2)^2 + (M_3-m_3)^2}$

*Centrosome - Cid distance*. The centrosome (CS) - Cid distance was calculated using Cid and CS coordinates.

CS – Cid distance: = $\sqrt{(x_1-x_2)^2 + (y_1-y_2)^2 + (z_1-z_2)^2}$

Where $x_1, y_1, z_1$ correspond to CS and $x_2, y_2, z_2$ to Cid coordinates, respectively. Plotted values correspond to averaged values of all CS – Cid puncta distances and the corresponding standard deviations.

0 and 6 min corresponds to the appearance of the basal centrosome and 6 min thereafter.

'0' and '6' mins correspond to the appearance of the ectopic centrosome and 6 min thereafter.

*Cid – cid distance calculations*. The distance between Cid clusters was calculated using the coordinates of endogenous and ectopic Cid foci over multiple time points.

Endogenous to ectopic Cid: at least three endogenous Cid foci and one ectopic Cid foci were selected. The values were averaged.

Endogenous Cid to endogenous Cid: the distance between two endogenous Cid clusters were calculated using the formula above and averaged.

T was defined as the first timepoint where the distance between endogenous and ectopic cid fell below the highest observed inter-endogenous Cid distance. ΔT was defined as the time difference between nuclear envelope breakdown (NEB) and T. ΔT = T − NEB.

**Statistics and reproducibility**. Statistical analysis was performed using Graphpad prism 8. Statistical significance was determined using paired or unpaired t-test and one-way ANOVA. Significance was indicated as following: $*p < 0.05$, $**p < 0.01$, $***p < 0.001$, $****p < 0.0001$, ns; not significant. Exact $p$ values and complete statistical information can be found in Supplementary Data 1. The meta data used for the quantifications is compiled in the file Supplementary Data 2.

Measurements were taken from multiple distinct samples and from several independent experiments. Cell fusion experiments have been performed by 5 independent researchers with similar outcomes.

**Reporting summary**. Further information on research design is available in the Nature Research Reporting Summary linked to this article.

## Data availability

The authors declare that all data supporting the findings of this study are available within the paper and its supplementary files. Source data are available upon request.

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

## Acknowledgements

We thank Xin Chen, Cayetano Gonzalez, Roger Karess, Tomer Avidor-Reiss for fly stocks, David Salvador Garcia for generating the Pros::EGFP transgenic line, Sue Biggins and members of the Cabernard laboratory for helpful discussions and comments. This work was supported by the National Institutes of Health (1R01GM126029) and a Research Scholar grant from the American Cancer Society (130285-RSG-16253-01-CSM). Stocks obtained from the Bloomington Drosophila Stock Center (NIH P40OD018537) and from the Vienna Drosophila Resource Center (VDRC).

## Author contributions

This study was conceived by B.S., N.L., and C.C. C.R. provided some conceptual ideas early on. Technical feasibility was demonstrated by C.R. & C.C. B.S. and N.L. performed all the experiments with significant help from J.T. and R.C.S. B.S., N.L., J.T., R.C.S., and C.C. analyzed the data. B.S. and C.C. wrote the manuscript.

## Competing interests

The authors declare no competing interests.
