## [Peer Review File · Communications Biology]

Reviewers' comments:

Reviewer #1 (Remarks to the Author):

"Asymmetric chromatin retention and nuclear envelopes separate chromosomes in fused cells during mitosis", by Dr Cabernard and colleagues.

In this study, Sunchu et al., managed to generate Neuroblast/GMC fusion using laser ablation of cell/cell contacts in the *Drosophila* Central Nervous system to study mitotic spindle assembly. They found that depending on the cell cycle phase when fusion occurs, NB or GMC spindles do not assemble synchronously, chromatids are not congressed at the same speed but somehow the 2 spindle managed to assemble a bipolar spindle with 2 centrosomes at each pole. They also found nicely that NB centromeres occupy an apical positioning that is driven by undirect interaction with the Microtubules nucleated by the apical centrosome aster during interphase. In fused cells the presence of the apical aster delays chromosome congression and mitotic duration. This is accompanied by chromosome segregation errors and possibly SAC dependent mitotic delay when fusion is performed in prophase NBs.

This study was technically challenging and we acknowledge the quality of the experimental work, image acquisition and analyses. However, even if we feel that the result is interesting we are wondering if this paper is of sufficient interest for *Communication Biology* in its present form. The paper is complex mainly because several observations regarding spindle assembly, mitotic progression, spindle compartmentation, aster-dependent centromere positioning are not accurately documented or not clearly connected together.

In addition we noticed several shortcuts, forgotten references of published work and finally somehow vague and sometimes misleading conclusions. We also suggest changing the title that is possibly misleading as it suggests a mechanism that would lead to asymmetric chromatin segregation that has been shown in some other stem cell models. Actually what this story shows is that the whole centromeres are asymmetrically positioned within the NB nucleus and this asymmetry appears to negatively influence mitotic progression in NB/GMC fused cells and lead to chromosome segregation errors. The story would be complete if the authors were able to show that NB/GMC fusions would do better cell division and less errors when apical NB centromeres are symmetric (Ctb RNAi).

I also recommend to go straight to point and avoid too much complexity in the figures.

I do have specific concerns and questions:

1-The localization of the centromeric specific H3 variant (CENP-A homologue) Cid have already been described, please cite the appropriate references.

2-There is a shortcut between Cid asymmetric localization, chromatin localization, centromere positioning and kinetochores. Please state that Cid is used to label the position of the centromeres during interphase and will be the site where kinetochores are assembled during M phase for microtubule attachment and chromosome congression.

3-The notion of endogenous (NB) or exogenous (GMC) chromatin is subjective and I believe the term NB or GMC chromatin should be used instead.

4-I do have a problem with the concept "of chromatin mixing" page 9 (lane 251). This is over-interpretation. From the literature, the mitotic spindle is separated by membranes (lamin, until at least mid-metaphase, Roubinet et al., *Current Biology* 2021, and during cell division (CD8-GFP), Schweizer et al., 2015) and Figure 4r. There is therefore likely no mixing of chromosomes before this stage of cell division. However, no relevant images of membranes are provided in fused mitotic cells. It is therefore impossible to state about the compartmentation state of the GMC and NB spindles during cell division steps.

5-Figure 4 should be simplified as it is too complicated to read like it is.

6-The SAC should be assayed directly by using SAC component localization on kinetochores. We expect the GMC kinetochores to be labeled for longer periods with proteins such as Mad2.

7-Page 9, l 313. I don't think the grafting experiment could work. The original assay described the injection of whole pieces of brain to generate abdominal tumors that killed host flies. The authors mentioned here the injection of 2 to 6 damaged cells by brain (l345-347). Without the use of a positive control such as the injection of 2 to 6 brain tumor cells per fly abdomen (different than Brat RNAi whole brains) the assay and the conclusions are not valid. Given the difficulty of the assay, this part should be removed from the paper.

8- I feel that the authors should emphasize on the "position effect" of the GMC nuclei. In most cases, it is located in a basal position (far away from the NB centromeres) and this represents a strong bias. Please comment. I feel that spindle assembly is facilitated by loss of the apical aster in fusions but is not clearly stated in the manuscript.

9-To confirm their hypothesis, and given the strong expertise of the team in centrosome manipulation, I was wondering why sas-4 mutants that retain polarity but lack centrosomes (and MT-asters) were not used in this study. Alternatively, it would have been great to maintain centrosome nucleation of both centrosomes during interphase.

10-The fact that aneuploid NB do not proliferate has already been described by at least three groups please cite the appropriate studies.

11-There is no evidence for fusion or lack of fusion of metaphase plates from GMC and NB in this study, at least based on the presence of surrounding membranes. But adjacent spindles are found in many cases. Can the authors provide convincing images of the membranes (CD8-GFP that surrounds spindles during all step of division) with microtubules or DNA? The model in Figure 6d differs from the paper (l.354). The panel is too small and it is difficult to see the presence or absence of compartmentalization of metaphysical plates or mitotic spindles. In general, the explanatory diagrams are too small.

12-The Cis and the Trans notion compared to II and X spindles and brings more confusion (figure 2d and 2o).

13-Figure 3, panels b and c. Is it possible, for consistency to keep the apical centrosome at the top ? This is not a fusion, why using "endo" in the y axis ?

14-It would have been nice to use photo-convertible histone or Cid to follow chromosomes during mitosis to make more accurate conclusions regarding GMC and NB chromosome mixing. This is what movie 9 tries to show but the pseudocolor code used seems to me too subjective and unconvincing to conclude on the mixing of centromeres during mitosis.

15-It is unfortunate that the results on is study were not discussed and put in perspective with the observations made in the studies of DeLuca and Rieder. Please comment.

16-In movie 10, cytokinesis fails, and GMC DNA appears to not condense into chromosomes. Such poorly condensed GMC chromosomes are also detected in figure 2b (His2A, top right panel). Please comment on the possible effects caused by lagging chromosomes on cytokinesis efficiency.

Reviewer #2 (Remarks to the Author):

In this paper, the authors analyze the consequences of acute fusion between *Drosophila* NBs and GMCs. Their *in vivo* analysis focuses on the dynamics of Cid-labeled kinetochores and spindle formation in both unfused control NBs and NB-GMC "hybrid cells". The goal of this study is understanding the mechanisms underlying the mitotic behavior of NB and GMC nuclei in fused cells. I found the paper difficult to read both for the order of presentation of the experiments and for the excess of figures and schemes, many of which were not properly reported in the text. In addition, the authors should take care of several parts of the manuscripts that are unclear and even contradictory, and of some conclusions that are not fully supported by the data.

Major points

(1) The authors demonstrate that after fusion NBs and GMCs form independent spindles that often fuse in a single metaphase spindle. This is an expected result, suggested by several papers on fused mammalian cells, which also address the mechanisms underlying fusion between different spindles (see for example Gatlin et al., *Curr Biol* 19:287, 2009). The authors should clearly explain the difference between their *Drosophila* system and the other fused-cell systems so far analyzed.

(2) In Suppl. Fig. 5 the authors use the FUCCI system to document that the NB is surrounded by GMCs in different phases of the cell cycle. However, they do not explain what happens when a G1 GMC is fused to an S-G2 NB. Based on their results it seems that the GMC and the NB nuclei form a spindle and divide more or less at the same time. Does the NB nucleus wait until the GMC has fully replicated its DNA (as suggested by Figure 1b) or does the GMC undergo cell division with unreplicated or not completely replicated DNA? At page 11, (lines 305-312) they mention that most hybrid cells have segregation problems but the quantification reported in Fig. 5f is very confusing. In addition, in Fig. 5g they show that most hybrid cells form either heterokaryons or syncarions. Does this mean that most hybrid cells fail to complete cytokinesis? This contradicts what stated at page 5 (lines 98-100) that "most NB-GMC hybrid cells ... completed cytokinesis (Suppl. Fig. 1 b-e)". The authors should reconcile these conflicting conclusions. Perhaps, hybrid cells form a central spindle and a midbody but fail to complete abscission. If this is the case, it should be clearly stated in the text and properly discussed.

To improve the readability of the paper I further suggest moving the parts of the paper concerning the spindle assembly checkpoint (lines 274-292 and Fig 5) and chromosome segregation in hybrid cells (lines 295-312 and Fig. 5) immediately after the part that describes dual spindles formation (Figures 1 and 2).

(3) Although one subsection of the article is entitled "Ectopic spindles are nucleated from GMC centrosomes", the data reported in Figures 2m and supplementary Figure 2 do not clearly show that formation of the GMC spindle is mediated by the GMC centrosomes. Actually, it is not even clearly documented that spindle poles of GMCs are associated with an Asl-GFP signal. The hybrid cell of Figure 2m shows some Asl-GFP signals but does not show the Cherry-Jupiter labeled spindles. Supplementary Figure 2 shows the spindles MTs (possibly associated with Cherry-Jupiter- this is not specified in the legend) but does not show the centrosomes at the spindle poles. In this figure, only the presumptive position of the centrosome is indicated with arrowheads. To clarify the role of centrosomes in dual spindles formation the authors should examine hybrid cells from either *asl* or *Sas-4* mutants, which are devoid of functional centrosomes. These experiments would definitely demonstrate whether centrosomes are essential for twin spindle formation in hybrid cells, or whether these cells can form acentrosomal dual spindles. The latter possibility is quite likely, as *asl* or *Sas-4* mutant larvae have the ability to form functional anastral spindles and to survive to adulthood.

(4) A substantial part of the paper is devoted to the analysis of Cid-marked centromere behavior in both normal NBs and hybrid NB-GMC cells. This is an interesting aspect of NB biology but is only marginally related to the formation and behavior of dual spindles. The authors show that in normal NBs the centromeres are clustered near the apical pole of the cell and that this clustering is disrupted by colcemid treatment, suggesting that it is MT dependent. They further show that centromere clustering is disrupted by RNAi against centrobins (*cnb*) (but do not provide any molecular evidence on the efficiency of RNAi). Since *cnb* specifically controls the apical MTOC

activity during interphase, they conclude that polarized centromere localization in NBs depends on interphase MTs. They also stated that they failed to see MTs from the apical MTOC penetrating the nuclear envelope before prometaphase and suggested that "CID localization could either be tightly coordinated, or indirectly connected with apical MTOC microtubules". The data presented in this paper (quality of images and intensity of MT staining) are not sufficiently cogent to support the conclusion that all *cnb*-dependent MTs fail to penetrate the nuclear envelope. Perhaps, there are studies that support this inference. If so, they should be cited and properly discussed. Alternatively, the authors should modify their conclusions about the role and behavior of *cnb*-dependent MTs.

The data in wild type hybrid cells clearly show that the apical, endogenous MTOC retains endogenous Cid close to the apical cortex. However, the images of *cnb* RNAi cells are not equally clear. Particularly, video 7 is very confused and should be substituted with a better one.

(5) The authors should provide a clear description of the transplantation experiments. They should specify the approximate number of cells injected and the number of fused cells in each injected sample. In addition, the images in Figure 6a are quite poor and it is hard to understand that they represent transplanted adult females. Furthermore, they should clearly state that they do not know whether polyploid hybrid cells can survive in transplants, and discuss this point in relation to tumor formation.

Additional comments on Figures and Videos

Fig. 1 The schemes in panels (e) and (f) should be briefly described. The sentence "Colored boxes represent corresponding cell cycle stages." should be moved at the end of panel (b) description, to facilitate reader's understanding. The authors should also address, either in the text or in the legend, why GMC chromatin is less fluorescent than NB chromatin.

Fig. 2 In panel (b) it is not clear how authors can assess that the 00:00 timing corresponds to NEB, as no nuclear envelope marker is shown and the colored box indicates that the cell is in interphase.

In panel (e) frequencies are not correctly reported as their sum does not correspond to 100. In the description of panel (g) authors refer to plotted data in (g) but the data are instead plotted in (h).

Based on the pictures in panel (m) I do not see how the endo or ecto origin of the centrosome signals can be assessed. This panel should be either substituted with a panel showing both centrosomes and MTs or removed.

Panel (n) is rather obvious and does not need statistical analysis.

Panels (p) and (q) are not cited in the main text and panel (q) is not mentioned in the legend.

Again, the sentence "Colored boxes represent corresponding cell cycle stages." should be moved at the end of panel (b) description, to facilitate reader's understanding.

Fig. 3 In panels (a,c,d,f) what does time 0 correspond to?

The CS abbreviation (centrosome) should be explained the first time it is used (panel b) and not in the following figure 4. By the way, in these pictures what they indicate as CS is not a centrosome but a MT aster.

Overall, the description of panels (e, f) is rather confusing. I do not understand where the orange cross and the closed arrows are.

Fig. 4 In panels (a) and (b) His2A should be substituted with Cid.

In panel (q) the meaning of Δt and of the horizontal dashed grey line should be briefly explained.

In panel (r) the images in the second row are not described.

Fig. 5 In panel (b) the quality of the images is poor and the dynamics of the endo and ecto chromatin is very difficult to interpret. In particular, it is very difficult to assess whether bridges are formed by endo or ecto chromosomes.

The description of panel (d) ("Average time difference...", line 781) is not precise, since it is a

scatter plot showing the median time difference.

Fig. 6 The model depicted in panel (d) seems rather arbitrary, since I do not see how ectopic chromosomes can be unambiguously distinguished from endo chromosomes within the chromatin clouds shown in Figures 5b and 5e. In addition, the final outcome (two distinct nuclei of different sizes) should be extremely rare, according to what reported in Figs (5f) and (5g).

Suppl. Fig. 1 In panel (a) what is the meaning of the different colors of GMCs?
In panel (c) I do not succeed to envisage any completion of cytokinesis.
In panel (e) there is a blue arrowhead that is not described in the legend.

Suppl. Fig. 2 Pictures in panel (a) are quite out of focus and nevertheless the dynamics of the endo and ecto spindles is much more understandable than in panel (b), where I find very difficult to discriminate between endo and ecto centrosomes once they have all matured. Here again, authors are visualizing MT asters and not centrosomes.
The method used to visualize MTs (cherry::Jupiter expression?) should be mentioned.

Suppl. Fig. 3 In panel (b) what does time 0:00 correspond to?
To facilitate the interpretation of what happens in fused cells, the authors should add a panel describing the behaviour of kinetochores in unperturbed GMCs.

Suppl. Fig. 4 Panels (a-d): they are scatter plots showing the median, not "Averaged distance..."
In panels (f) and (h) there is a blue asterisk which is not explained.
There are no colored boxes in this Figure.

Movies 5 and 6 The marker mCherry::CAAX is not mentioned in the videos

Reviewer #3 (Remarks to the Author):

The study establishes *Drosophila* neuroblasts as a model system to study the mechanisms behind the formation of dual spindles after an induced cell fusion event. The study finds that dual spindles form in neuroblasts that were induced to fuse with their daughter cells. The manuscript then explores potential mechanisms explaining this remarkable phenomenon addressing the important question of how cells can distinguish chromatin from different origins.

Neuroblasts have an unusual centrosome cycle and configuration of the microtubule network which is organised from the apical cortex in interphase. The study finds that an active MTOC and microtubules in interphase are required to keep centromeres orientated towards the apical pole. For dual spindle formation the study proposes that this connection contributes to preventing chromatin mixing upon fusion, and finally experiments are presented aiming at addressing if cell fusions are tumorigenic.

The question addressed here is of broad interest and timely. It is further a refreshing and creative use of this model to address a fundamental question. The study contains a range of interesting and report-worthy observations. The independent kinetics of alignment and spindle dynamics in the hybrids is very interesting for instance.

Apart from the finding that dual spindles can form and can have different kinetics upon fusion and the observation that centromeres are apically positioned in the neuroblast nucleus in interphase and depend on microtubules and the activity of the MTOC there is too little functional testing. I enjoyed the introduction and question setting, but then the manuscript appears quite descriptive and speculative, unfortunately the only functional analysis (*cnb*-RNAi) does not unambiguously demonstrate a role for the MTOC-centromere axis in preventing mixing of chromosomes of different origins upon fusion, especially given the complexity of the arrangement of elements measured without specific labels.

Suggestions for improvement

1) Tracing NB and GMC chromatin and organelles: The lack of labelling of the individual components traced leave room for alternative interpretations. While tracing of DNA based on Histon2A may allow distinguishing chromatin of different origins based on labelling thresholds, the situation becomes much more complex for tracing Cid. Precise labelling strategies would be important for stringent measurements. The lab has a track record of successful photolabeling approaches, could that help? Eos-Cid? Or established tools, the photoconvertible histone variants from the Chen lab?

2) Timing of fusion induction: The most compelling data is the formation of dual spindles ("II" type) in control hybrids. In this type spindles, centrosomes and chromatin look clearly separated. In X this seems more complex (provide a clear video of this type to discern from multipolar spindles). Is formation of II and X spindle types related to timing of fusion, e.g. are II types induced predominantly in mitotic neuroblasts when fused (as in video 2) and X in interphase? This is important for the interpretation.

Videos 8 and 9 further look like multipolar spindles and not dual or merged. If type II never forms in *cnb*-RNAi, then what is the interpretation? Movies of type II dual spindles in *cnb*-RNAi with Tubulin and Cid would be helpful.

3) Potentially confounding factors for *cnb*-RNAi interpretation: Even if chromatin mixing was a result, which is possible, the experimental design does not unambiguously allow assigning this to changes in the proposed Cid-centrosome connection. What about differences in spindle formation kinetics (e.g. Fig 3 a,d)? Slower microtubule dynamics could affect the ability to form dual, mixed, multipolar spindles. It is important to correlate the Cid patterns in *cnb*-RNAi with the type of spindle in the cell analysed for a cleaner interpretation. Position of the cuts to induce fusion: How can it be assured that the angle at which the foreign nucleus and its organelles enter the neuroblasts are not relevant to the process, especially when fusion is induced in mitosis? This might be related to the *cnb*-RNAi idea.

4) The paragraph line 218 is not clear. The title is about the role of the apical MTOC on chromosome mixing and implies that this can be experimentally perturbed. The finding then is ectopic centrosomes are closer to ectopic Cid upon *cnb*-RNAi. What does that mean? It reads like they remain together, hence "unmixed"? The data seems to measure that without the MTOC the inner neuroblast organisation in terms of Cid-mitotic centrosome distance is altered, yet own and foreign are distinguished? The logic of why that allows suggesting that chromosomes are mixed in *cnb*-RNAi is not clear.

5) The study evokes "asymmetric chromatin-centrosome connections". The connection between the chromosomes in the nucleus appears to be rather indirect (as the authors point out). Another interpretation: the apical centrosome and resulting microtubule network hold the nucleus in place and shape, which is perturbed if you lose the interphase MTOC. Consequently, in *cnb*-RNAi nuclear position and shape change and the nucleus might rotate (and perhaps changes in the distribution of other organelles) result. Can this be ruled out to contribute?

6) There are other ways to bring about loss of interphase MTOC activity in neuroblasts that could be tested.

7) The nuclear envelope analysis is descriptive and surprisingly not tested on *cnb*-RNAi fusion hybrids.

Other

Does dual spindle formation require fused cells to be in the cell cycle? What about apical cells that withdraw from the cell cycle (according to FUCCI data) are they competent to form spindles when placed in a mitotic cytoplasm?

Videos: Does cytokinesis take place normally upon fusion? None of the videos actually show that. This is important for the analysis of lagging chromosomes and the tumorigenicity test.

Fig 6, this experiment, while interesting is not adding much. The sample size is likely to be too low to rule out that the fusion induces tumours. Have Brat controls been imaged and laser cut with unsuccessful fusion induction?

The quite specific configuration of neuroblasts centrosome regulation raise the question of transferability.

Fig 1 provide images of spindles resulting from fusions induced in interphase, prophase, metaphase...

Fig 2d provide images on how X and II actually look like

Fig 4 the label is wrong this is Cid in blue not Histone 2A

Line 145 clarify to what "formed at the same time refers to"

Line 132 paragraph poses the question whether single or multiple spindles underly independent chromatin alignment at the beginning and finishes by concluding that spindle morphology is not important. Confusing, perhaps rewrite?

Check methods some detail not visible.

Line 168 "centrosome clustering" is an unlucky term. It suggests that centrosomes are found next to each other, while, if I got it right the fact that the NBs centrosomes are found in one spindle and the GMC centrosomes are found in the other is meant. Perhaps reword?

Information about the time resolution of imaging and when the cuts are made is critical, but missing for the experiments.

The sample size is rather small in places.

The criteria for choosing the site of cuts (I guess heterochromatin to judge apical) should be clearly stated and the site of cuts indicated in the relevant figures.

Reviewers's comments:

Reviewer #1 (Remarks to the Author):

"Asymmetric chromatin retention and nuclear envelopes separate chromosomes in fused cells during mitosis", by Dr Cabernard and colleagues.

In this study, Sunchu et al., managed to generate Neuroblast/GMC fusion using laser ablation of cell/cell contacts in the Drosophila Central Nervous system to study mitotic spindle assembly. They found that depending on the cell cycle phase when fusion occurs, NB or GMC spindles do not assemble synchronously, chromatids are not congressed at the same speed but somehow the 2 spindle managed to assemble a bipolar spindle with 2 centrosomes at each pole. They also found nicely that NB centromeres occupy an apical positioning that is driven by undirect interaction with the Microtubules nucleated by the apical centrosome aster during interphase. In fused cells the presence of the apical aster delays chromosome congression and mitotic duration. This is accompanied by chromosome segregation errors and possibly SAC dependent mitotic delay when fusion is performed in prophase NBs.

This study was technically challenging and we acknowledge the quality of the experimental work, image acquisition and analyses. However, even if we feel that the result is interesting we are wondering if this paper is of sufficient interest for Communication Biology in its present form. The paper is complex mainly because several observations regarding spindle assembly, mitotic progression, spindle compartmentation, aster-dependent centromere positioning are not accurately documented or not clearly connected together.

In addition we noticed several shortcuts, forgotten references of published work and finally somehow vague and sometimes misleading conclusions. We also suggest changing the title that is possibly misleading as it suggests a mechanism that would lead to asymmetric chromatin segregation that has been shown in some other stem cell models. Actually what this story shows is that the whole centromeres are asymmetrically positioned within the NB nucleus and **this asymmetry appears to negatively influence mitotic progression in NB/GMC fused cells** and lead to chromosome segregation errors. The story would be complete if the authors were able to show that NB/GMC fusions would do better cell division and less errors when apical NB centromeres are symmetric (Ctb RNAi).

I also recommend to go straight to point and avoid too much complexity in the figures.

I do have specific concerns and questions:

1-The localization of the centromeric specific H3 variant (CENP-A homologue) Cid have already been described, please cite the appropriate references.

We already cited the original paper from Steven Henikoff (Henikoff et al., 2000) and the recent paper from the Chen lab (Rajan et al., 2019). We added the Dattoli et al., reference. If the reviewer feels there are other references that should be cited, we would be happy to include these. To our knowledge, asymmetric Cid localization in fly neuroblasts has not been shown so far.

2-There is a shortcut between Cid asymmetric localization, chromatin localization, centromere positioning and kinetochores. Please state that Cid is used to label the position of the centromeres during interphase and will be the site where kinetochores are assembled during M phase for microtubule attachment and chromosome congression.

We have changed the manuscript as follows: “The centromere-specific H3 variant (Centromere identifier (Cid) in flies) colocalizes with centromeres ¹. Since sister chromatids in *Drosophila* male germline stem cells contain asymmetric levels of Cid ^{2,3}, we investigated whether hybrid cell spindles differentiate between endogenous and ectopic chromosomes based on differing levels of Cid.”

We believe this statement clarifies the motivation for using Cid as a marker in our study.

1. Henikoff, S., Ahmad, K., Platero, J. S. & Steensel, B. van. Heterochromatic deposition of centromeric histone H3-like proteins. *Proc National Acad Sci* 97, 716–721 (2000).

2. Ranjan, R., Snedeker, J. & Chen, X. Asymmetric Centromeres Differentially Coordinate with Mitotic Machinery to Ensure Biased Sister Chromatid Segregation in Germline Stem Cells. *Cell Stem Cell* (2019) doi:10.1016/j.stem.2019.08.014.

3. Dattoli, A. A. *et al.* Asymmetric assembly of centromeres epigenetically regulates stem cell fate. *J Cell Biology* 219, (2020).

3-The notion of endogenous (NB) or exogenous (GMC) chromatin is subjective and I believe the term NB or GMC chromatin should be used instead.

We have modified the manuscript in several places to refer specifically to NB and GMC chromatin.

4-I do have a problem with the concept “of chromatin mixing” page 9 (lane 251). This is over-interpretation. From the literature, the mitotic spindle is separated by membranes (lamin, until at least mid–metaphase, Roubinet *et al.*, *Current Biology* 2021, and during cell division (CD8-GFP), Schweizer *et al.*, 2015) and Figure 4r. There is therefore likely no mixing of chromosomes before this stage of cell division. However, no relevant images of membranes are provided in fused mitotic cells. It is therefore impossible to state about the compartmentation state of the GMC and NB spindles during cell division steps.

We added new data of fusion cells showing Lamin::GFP and cherry::Jupiter. As Figure 6b shows, the GMC and NB nuclear envelope are clearly discernable until early metaphase when the two spindles are oriented in parallel to each other. From that point onward, the NE signal is starting to diffuse and the separation between NB NE and GMC NE are not so clear anymore.

We have changed the wording in the manuscript to avoid an overinterpretation of our data.

5-Figure 4 should be simplified as it is too complicated to read like it is.

We have simplified Figure 4 and moved some of the data into the supplemental figures and made an additional figure.

6-The SAC should be assayed directly by using SAC component localization on kinetochores. We expect the GMC kinetochores to be labeled for longer periods with proteins such as Mad2.

We analyzed hybrid cells expressing Mad2::GFP. Indeed, some hybrid cells show Mad2 signal persisting on GMC kinetochores after it faded on NB kinetochores (e.g. supplementary Figure 4d). In some cases, Mad2::GFP consolidates in a single spot though (Figure 1h & supplementary Figure 4c). We think this is consistent with the data shown in Figure 1f, g.

7-Page 9, l 313. I don't think the grafting experiment could work. The original assay described the injection of whole pieces of brain to generate abdominal tumors that killed host flies. The authors mentioned here the injection of 2 to 6 damaged cells by brain (l345-347). Without the use of a positive control such as the injection of 2 to 6 brain tumor cells per fly abdomen (different than Brat RNAi whole brains) the assay and the conclusions are not valid. Given the difficulty of the assay, this part should be removed from the paper.

This is a fair point. Although we were cautious with the interpretation of these transplantation experiments, we ultimately decided to remove it from the manuscript.

8- I feel that the authors should emphasize on the "position effect" of the GMC nuclei. In most cases, it is located in a basal position (far away from the NB centromeres) and this represents a strong bias. Please comment. I feel that spindle assembly is facilitated by loss of the apical aster in fusions but is not clearly stated in the manuscript.

Position effect: this is a good point. GMCs are positioned basally because the mitotic spindle maintains its position for several divisions. Also, cortex glia surrounds the parental NB and its progeny, which provides an additional cell layer on the apical NB side. In other words, any GMCs in contact with a NB on the apical side are separated by another cell layer. Thus, we opted not to fuse apically located GMCs.

We added a brief section to the discussion to address this point.

9-To confirm their hypothesis, and given the strong expertise of the team in centrosome manipulation, I was wondering why sas-4 mutants that retain polarity but lack centrosomes (and MT-asters) were not used in this study. Alternatively, it would have been great to maintain centrosome nucleation of both centrosomes during interphase.

Done. We included data of *as*/ hybrid cells (lacking functional centrosomes throughout the cell cycle) and CnbPACT hybrids which contain two active interphase MTOCs. The new data is shown in Figure 3g, h; Figure 5h-j and supplementary figure 7.

10-The fact that aneuploid NB do not proliferate has already been described by at least three groups please cite the appropriate studies.

Because we removed the transplantation experiments from the manuscript, this paragraph has also been removed.

11-There is no evidence for fusion or lack of fusion of metaphase plates from GMC and NB in this study, at least based on the presence of surrounding membranes. But adjacent spindles are found in many cases. Can the authors provide convincing images of the membranes (CD8-GFP that surrounds spindles during all step of division) with microtubules or DNA? The model in Figure 6d differs from the paper (l.354). The panel is too small and it is difficult to see the presence or absence of compartmentalization of metaphysical plates or mitotic spindles. In general, the explanatory diagrams are too small.

We increased the size of all schematics. Also, as mentioned above, we included new LaminGFP hybrid cell data (Figure 6b), which shows that the NE becomes very diffuse during late metaphase. We cannot unambiguously state whether NB and GMC chromatin are still separated by membranes.

12-The Cis and the Trans notion compared to II and X spindles and brings more confusion (figure 2d and 2o).

We tried to clarify this concept in the revised manuscript.

13-Figure 3, panels b and c. Is it possible, for consistency to keep the apical centrosome at the top ? This is not a fusion, why using "endo" in the y axis ?

The apical centrosome is kept on top, both in the schematics and images, unless it rotates such as in *cnb* RNAi. We kept the 'endo' term in the y axis to make it consistent with the other graphs.

14-It would have been nice to use photo-convertible histone or Cid to follow chromosomes during mitosis to make more accurate conclusions regarding GMC and NB chromosome mixing. This is what movie 9 tries to show but the pseudocolor code used seems to me too subjective and unconvincing to conclude on the mixing of centromeres during mitosis.

Indeed! It was our dream experiment and we tried very hard to get it to work. Unfortunately, the signal-to-noise was too low and the conversion efficiency was not 100%. So, we were unable to obtain conclusive data.

We have included the original movie with the pseudocolor movie so the readers can make their own conclusions. We carefully and conservatively tracked Cid clusters and centrosomes in these movies so we are confident that the pseudocolors represent the actual situation.

15-It is unfortunate that the results on this study were not discussed and put in perspective with the observations made in the studies of DeLuca and Rieder. Please comment.

We have added a brief discussion to the manuscript. However, we will point out that the focus of our study differs from these two papers.

16-In movie 10, cytokinesis fails, and GMC DNA appears to not condense into chromosomes. Such poorly condensed GMC chromosomes are also detected in figure 2b (His2A, top right panel). Please comment on the possible effects caused by lagging chromosomes on cytokinesis efficiency.

We think that lagging chromosomes do not necessarily affect cytokinesis in all cases. For instance, the new Figure 7c shows an example of a hybrid cell with lagging chromosomes forming a synkaryon. In this case, cytokinesis appears to complete normally, judging from the robust midbody that is formed (see supplementary figure 8). Also, we see other examples of lagging chromosomes that do not affect cytokinesis (e.g supplementary figure 3c, d). Although we cannot exclude that some hybrid cells fail to complete cytokinesis, we generally do not see this happening in our data.

Reviewer #2 (Remarks to the Author):

In this paper, the authors analyze the consequences of acute fusion between *Drosophila* NBs and GMCs.

Their in vivo analysis focuses on the dynamics of Cid-labeled kinetochores and spindle formation in both unfused control NBs and NB-GMC "hybrid cells". The goal of this study is understanding the mechanisms underlying the mitotic behavior of NB and GMC nuclei in fused cells. I found the paper difficult to read both for the order of presentation of the experiments and for the excess of figures and schemes, many of which were not properly reported in the text. In addition, the authors should take care of several parts of the manuscripts that are unclear and even contradictory, and of some conclusions that are not fully supported by the data.

Major points

(1) The authors demonstrate that after fusion NBs and GMCs form independent spindles that often fuse in a single metaphase spindle. This is an expected result, suggested by several papers on fused mammalian cells, which also address the mechanisms underlying fusion between different spindles (see for example Gatlin et al., Curr Biol 19:287, 2009). The authors should clearly explain the difference between their *Drosophila* system and the other fused-cell systems so far analyzed.

We do not necessarily think that this result is expected. To our knowledge, all previous fusion experiments have either been performed with cells of the same type in culture or analyzed the first division after fertilization. Thus, our system differs in the sense that we are analyzing chromosome and spindle dynamics in vivo between molecularly distinct cell types. We tried to reinforce that point in the revised introduction and discussion.

(2) In Suppl. Fig. 5 the authors use the FUCCI system to document that the NB is surrounded by GMCs in different phases of the cell cycle. However, they do not explain what happens when a G1 GMC is fused to an S-G2 NB. Based on their results it seems that the GMC and the NB nuclei form a spindle and divide more or less at the same time. Does the NB nucleus wait until the GMC has fully replicated its DNA (as suggested by Figure 1b) or does the GMC undergo cell division with unreplicated or not completely replicated DNA? At page 11, (lines 305-312) they mention that most hybrid cells have segregation problems but the quantification reported in Fig. 5f is very confusing. In addition, in Fig. 5g they show that most hybrid cells form either heterokaryons or syncarions. Does this mean that most hybrid cells fail to complete cytokinesis? This contradicts what stated at page 5 (lines 98-100) that "most NB-GMC hybrid cells ... completed cytokinesis (Suppl. Fig. 1 b-e)".

The authors should reconcile these conflicting conclusions. Perhaps, hybrid cells form a central spindle and a midbody but fail to complete abscission. If this is the case, it should be clearly stated in the text and properly discussed.

In the revised manuscript, we included data of a hybrid cell, derived from a fusion between a mitotic NB and a G1-S GMC (supplementary figure 2. Needless to say, this data is not very easy to come by). We observed, consistent with the data shown previously, that the GMC quickly adjusts its cell cycle to the NB. We also included data showing hybrid cells expressing Mad2::GFP and confirmed our previous findings that either show GMC chromatin separating at the same time with NB chromatin or with a delay. There is no doubt that some fusions occurred between NBs and GMCs with unreplicated chromatin. But our data suggests that even GMC nuclei with unreplicated chromatin appear to adjust to the NB cell cycle.

We also reanalyzed our movies in regards of cytokinesis failure. The aforementioned syncaryon is a result of a successfully completed cytokinesis, based on the formation of a robust midbody as shown in supplementary figure 8. We think that in the syncaryon shown in figure 7 (formerly figure 5), the two nuclei merge into one.

Other examples, where chromosomes are lagging also seem to complete cytokinesis. See supplementary figure 3c,d,e.

To improve the readability of the paper I further suggest moving the parts of the paper concerning the spindle assembly checkpoint (lines 274-292 and Fig 5) and chromosome segregation in hybrid cells (lines 295-312 and Fig. 5) immediately after the part that describes dual spindles formation (Figures 1 and 2).

We thank the reviewer for this suggestion. We reorganized the revised manuscript accordingly.

(3) Although one subsection of the article is entitled “Ectopic spindles are nucleated from GMC centrosomes”, the data reported in Figures 2m and supplementary Figure 2 do not clearly show that formation of the GMC spindle is mediated by the GMC centrosomes. Actually, it is not even clearly documented that spindle poles of GMCs are associated with an Asl-GFP signal. The hybrid cell of Figure 2m shows some Asl-GFP signals but does not show the Cherry-Jupiter labeled spindles. Supplementary Figure 2 shows the spindles MTs (possibly associated with Cherry-Jupiter- this is not specified in the legend) but does not show the centrosomes at the spindle poles. In this figure, only the presumptive position of the centrosome is indicated with arrowheads. To clarify the role of centrosomes in dual spindles formation the authors should examine hybrid cells from either *asl* or *Sas-4* mutants, which are devoid of functional centrosomes. These experiments would definitely demonstrate whether centrosomes are essential for twin spindle formation in hybrid cells, or whether these cells can form acentrosomal dual spindles. The latter possibility is quite likely, as *asl* or *Sas-4* mutant larvae have the ability to form functional anastral spindles and to survive to adulthood.

We modified the *Asl::GFP* figure and included better images in the new figure 3a,b. It clearly shows that in hybrid cells *Asl::GFP* and *cherry::Jupiter* overlap at maturing centrosomes.

We also performed fusions in *asl* mutant brains. The problem is that *asl* mutant NBs can form unfocused mitotic spindles as reported before (and now shown in Figure 3h). Indeed, *asl* mutant hybrid cells are able to form mitotic spindles but it is not discernable whether they form dual spindles or a single spindle. At any rate, these hybrid cells are still separating GMC and NB chromatin in early mitosis and congress it independently of each other at the metaphase plate as shown in figure 3h.

(4) A substantial part of the paper is devoted to the analysis of Cid-marked centromere behavior in both normal NBs and hybrid NB-GMC cells. This is an interesting aspect of NB biology but is only marginally related to the formation and behavior of dual spindles. The authors show that in normal NBs the centromeres are clustered near the apical pole of the cell and that this clustering is disrupted by colcemid treatment, suggesting that it is MT dependent. They further show that centromere clustering is disrupted by RNAi against centrobin (*cnb*) (but do not provide any molecular evidence on the efficiency of RNAi). Since *cnb* specifically controls the apical MTOC activity during interphase, they conclude that polarized centromere localization in NBs depends on interphase MTs. They also stated that they failed to see MTs from the apical MTOC penetrating the nuclear envelope before prometaphase and suggested that “CID localization could either be tightly coordinated, or indirectly connected with apical MTOC microtubules”. The data presented in this paper (quality of images and intensity of MT staining) are not sufficiently cogent to support the conclusion that all *cnb*-dependent MTs fail to penetrate the nuclear envelope. Perhaps, there are studies that support this inference. If so, they should be cited and properly discussed. Alternatively, the authors should modify their conclusions about the role and behavior of *cnb*-dependent MTs.

Januscke et al., reported that the *cnb* RNAi shows the same phenotype as the *cnb*^{e00267} loss of function allele crossed over a *cnb* deficiency chromosome. We have independently confirmed this in previous experiments. Also, loss of MTs causes the apical centriole to lose contact with the apical NB cortex, which is exactly what we see in the *cnb* RNAi. The *cnb* RNAi, colcemid and our imaging experiments combined suggest that apical Cid localization is dependent on the apical MTOC.

We think the connection between the apical MTOC and Cid localization is quite critical for the manuscript, as it could explain the separation between NB and GMC chromatin.

We have modified the manuscript to account for the possibility that MTs penetrate the NE before mitosis.

The data in wild type hybrid cells clearly show that the apical, endogenous MTOC retains endogenous

Cid close to the apical cortex. However, the images of *cnb* RNAi cells are not equally clear. Particularly, video 7 is very confused and should be substituted with a better one.

We replaced video 7 with a new *cnb* RNAi movie and also changed the corresponding image sequence and plot in Figure 4.

(5) The authors should provide a clear description of the transplantation experiments. They should specify the approximate number of cells injected and the number of fused cells in each injected sample. In addition, the images in Figure 6a are quite poor and it is hard to understand that they represent transplanted adult females. Furthermore, they should clearly state that they do not know whether polyploid hybrid cells can survive in transplants, and discuss this point in relation to tumor formation.

We removed this data from the manuscript.

Additional comments on Figures and Videos

Fig. 1 The schemes in panels (e) and (f) should be briefly described. The sentence "Colored boxes represent corresponding cell cycle stages." should be moved at the end of panel (b) description, to facilitate reader's understanding. The authors should also address, either in the text or in the legend, why GMC chromatin is less fluorescent than NB chromatin.

We changed the legend accordingly. Because we can only speculate about the intensity differences between NB and GMC, we did not include this in the manuscript.

Fig. 2 In panel (b) it is not clear how authors can assess that the 00:00 timing corresponds to NEB, as no nuclear envelope marker is shown and the colored box indicates that the cell is in interphase.

This panel indicates the start of NEB, which we assess based on changes in His2A::GFP localization (see below). We have added the label NEB to this image.

In panel (e) frequencies are not correctly reported as their sum does not correspond to 100.

This graph is updated and corrected.

In the description of panel (g) authors refer to plotted data in (g) but the data are instead plotted in (h).

Corrected.

Based on the pictures in panel (m) I do not see how the endo or ecto origin of the centrosome signals can be assessed. This panel should be either substituted with a panel showing both centrosomes and MTs or removed.

Panel (n) is rather obvious and does not need statistical analysis.

Panels (p) and (q) are not cited in the main text and panel (q) is not mentioned in the legend.

Again, the sentence "Colored boxes represent corresponding cell cycle stages." should be moved at the end of panel (b) description, to facilitate reader's understanding.

We made a new figure 3 with this data. It is now referenced in the text and the legend has been updated.

Fig. 3 In panels (a,c,d,f) what does time 0 correspond to?

The CS abbreviation (centrosome) should be explained the first time it is used (panel b) and not in the following figure 4. By the way, in these pictures what they indicate as CS is not a centrosome but a MT aster.

Overall, the description of panels (e, f) is rather confusing. I do not understand where the orange cross and the closed arrows are.

We have updated the legend as best as we could. The legend defines the abbreviation CS and also explains that 0 corresponds to the time when the basal centrosome appears. The main text also contains this information. The orange cross, closed and open arrows are all explained in the figure and legend.

Fig. 4 In panels (a) and (b) His2A should be substituted with Cid.

In panel (q) the meaning of Δt and of the horizontal dashed grey line should be briefly explained.

In panel (r) the images in the second row are not described.

We corrected and updated this figure. It should be more intuitive and easier to understand now.

Fig. 5 In panel (b) the quality of the images is poor and the dynamics of the endo and ecto chromatin is very difficult to interpret. In particular, it is very difficult to assess whether bridges are formed by endo or ecto chromosomes.

The description of panel (d) ("Average time difference...", line 781) is not precise, since it is a scatter plot showing the median time difference.

We corrected the figure legend. Note that this data is now moved to Figure 1.

Fig. 6 The model depicted in panel (d) seems rather arbitrary, since I do not see how ectopic chromosomes can be unambiguously distinguished from endo chromosomes within the chromatin clouds shown in Figures 5b and 5e. In addition, the final outcome (two distinct nuclei of different sizes) should be extremely rare, according to what reported in Figs (5f) and (5g).

We have updated the model in the revised Figure 7. Some aspects remain unclear, which is indicated in the model. The distinction of endo and ectopic chromosomes from metaphase onward is based on the observation that endo and ecto chromosomes can be distinguished based on intensity differences. Furthermore, we have documented that ectopic chromosomes can segregate with a delay compared to endogenous chromosomes. This implies that ectopic chromatin is still attached to the GMC-derived mitotic spindle, which independently drives GMC chromatid separation.

Suppl. Fig. 1 In panel (a) what is the meaning of the different colors of GMCs?

In panel (c) I do not succeed to envisage any completion of cytokinesis.

In panel (e) there is a blue arrowhead that is not described in the legend.

The figure legend has been corrected.

These hybrid cells express Myosin (Sqh::GFP), which clearly shows that cytokinesis completes.

Suppl. Fig. 2 Pictures in panel (a) are quite out of focus and nevertheless the dynamics of the endo and ecto spindles is much more understandable than in panel (b), where I find very difficult to discriminate between endo and ecto centrosomes once they have all matured. Here again, authors are visualizing MT

asters and not centrosomes.

The method used to visualize MTs (cherry::Jupiter expression?) should be mentioned.

We updated the figure legends. Cherry::Jupiter was used and mentioned now. The images are not perfectly in focus as this NB divided within the Z-plane. Centrosomes/MTOCs (were traced in the original movie frame by frame and we are confident that our representation accurately reflects the distribution of the MTOCs.

Suppl. Fig. 3 In panel (b) what does time 0:00 correspond to?

To facilitate the interpretation of what happens in fused cells, the authors should add a panel describing the behaviour of kinetochores in unperturbed GMCs.

We do not think showing the localization of Cid in GMCs is particularly informative for the reader. As the image below indicates, the localization is quite variable and currently subject to further investigation.

Suppl. Fig. 4 Panels (a-d): they are scatter plots showing the median, not “Averaged distance...”

In panels (f) and (h) there is a blue asterisk which is not explained.

There are no colored boxes in this Figure.

We updated the figure legends and replaced or removed the panels (f) and (h).

Movies 5 and 6 The marker mCherry::CAAX is not mentioned in the videos

Corrected.

Reviewer #3 (Remarks to the Author):

The study establishes *Drosophila* neuroblasts as a model system to study the mechanisms behind the formation of dual spindles after an induced cell fusion event. The study finds that dual spindles form in neuroblasts that were induced to fuse with their daughter cells. The manuscript then explores potential mechanisms explaining this remarkable phenomenon addressing the important question of how cells can distinguish chromatin from different origins.

Neuroblasts have an unusual centrosome cycle and configuration of the microtubule network which is organised from the apical cortex in interphase. The study finds that an active MTOC and microtubules in interphase are required to keep centromeres orientated towards the apical pole. For dual spindle formation the study proposes that this connection contributes to preventing chromatin mixing upon fusion, and finally experiments are presented aiming at addressing if cell fusions are tumorigenic.

The question addressed here is of broad interest and timely. It is further a refreshing and creative use of this model to address a fundamental question. The study contains a range of interesting and report-worthy observations. The independent kinetics of alignment and spindle dynamics in the hybrids is very interesting for instance.

Apart from the finding that dual spindles can form and can have different kinetics upon fusion and the observation that centromeres are apically positioned in the neuroblast nucleus in interphase and depend on microtubules and the activity of the MTOC there is too little functional testing. I enjoyed the introduction and question setting, but then the manuscript appears quite descriptive and speculative, unfortunately the only functional analysis (*cnb*-RNAi) does not unambiguously demonstrate a role for the MTOC-centromere axis in preventing mixing of chromosomes of different origins upon fusion, especially given the complexity of the arrangement of elements measured without specific labels.

Suggestions for improvement

1) Tracing NB and GMC chromatin and organelles: The lack of labelling of the individual components traced leave room for alternative interpretations. While tracing of DNA based on Histon2A may allow distinguishing chromatin of different origins based on labelling thresholds, the situation becomes much more complex for tracing Cid. Precise labelling strategies would be important for stringent measurements. The lab has a track record of successful photolabeling approaches, could that help? Eos-Cid? Or established tools, the photoconvertible histone variants from the Chen lab?

Using a photoconversion approach to distinguish between NB and GMC chromatin was our dream experiment and we tried very hard to get it to work. Unfortunately, the signal-to-noise was too low and the conversion efficiency was not 100%. We predominantly tried the photoconvertible Cid variant from the Chen lab but were unable to obtain conclusive data.

2) Timing of fusion induction: The most compelling data is the formation of dual spindles ("II" type) in control hybrids. In this type spindles, centrosomes and chromatin look clearly separated. In X this seems more complex (provide a clear video of this type to discern from multipolar spindles). Is formation of II and X spindle types related to timing of fusion, e.g. are II types induced predominantly in mitotic neuroblasts when fused (as in video 2) and X in interphase? This is important for the interpretation. Videos 8 and 9 further look like multipolar spindles and not dual or merged. If type II never forms in *cnb*-RNAi, then what is the interpretation? Movies of type II dual spindles in *cnb*-RNAi with Tubulin and Cid would be helpful.

We have performed additional analysis on this question. As already shown in the first version – now updated with more data – II vs X-type spindles are dependent on fusion induction. Although some exceptions occur, II spindles predominantly occur when fusions were induced closer to nuclear envelope breakdown (NEB). We provided additional movies to show the distinction between these types.

For *cnb* RNAi, we predominantly obtain X-type spindles, which could be explained by the fact that all centrosomes mature at the same time and thus have equal opportunity to pair with an NB or GMC centrosome. In *cnb* RNAi, the time between fusion induction and NEB does not have a significant influence on whether X or II type spindles are being formed. This data is shown in figure 5i,j.

3) Potentially confounding factors for *cnb*-RNAi interpretation: Even if chromatin mixing was a result, which is possible, the experimental design does not unambiguously allow assigning this to changes in the proposed Cid-centrosome connection. What about differences in spindle formation kinetics (e.g. Fig 3 a,d)? Slower microtubule dynamics could affect the ability to form dual, mixed, multipolar spindles. It is important to correlate the Cid patterns in *cnb*-RNAi with the type of spindle in the cell analysed for a cleaner interpretation. Position of the cuts to induce fusion: How can it be assured that the angle at which the foreign nucleus and its organelles enter the neuroblasts are not relevant to the process, especially when fusion is induced in mitosis? This might be related to the *cnb*-RNAi idea.

There is little evidence to suggest that microtubule dynamics differ between the apical and basal neuroblast centrosome. Previous studies – including from our lab – showed that the apical NB centrosome maintains MTOC activity during interphase whereas the basal centrosome only regains MTOC activity in early mitosis. At that point, both centrosomes seem to mature with equal dynamics.

cnb RNAi and conversely *cnb* PACT changes MTOC asymmetry: the former creates two inactive, the latter two active MTOCs during interphase. Thus, under these conditions, all centrosomes have the same activity. As mentioned above, we have made additional fusions in *cnb* RNAi mutant brains and obtain predominantly X-type spindles. Also, we generated hybrids in *cnb*-PACT expressing brains and also obtain more X-type spindles compared to wild type. Thus, we conclude that MTOC activity is correlated with spindle-architecture in hybrid cells and we included this analysis in figure 5h-j.

Position of the cuts: GMCs are predominantly located on the basal side of the NB, thus we have relatively little control over where to induce fusions. We accounted for this in the discussion. For the later part of the manuscript, we based our conclusions mostly on fusions induced in either interphase or early prophase so we think the angle is not the main factor for the observed observations. Again, if that would be the case, we would not see a significant change in GMC-NB Cid distance dynamics as measured in Figure 5h.

4) The paragraph line 218 is not clear. The title is about the role of the apical MTOC on chromosome mixing and implies that this can be experimentally perturbed. The finding then is ectopic centrosomes are closer to ectopic Cid upon *cnb*-RNAi. What does that mean? It reads like they remain together, hence “unmixed”? The data seems to measure that without the MTOC the inner neuroblast organisation in terms of Cid-mitotic centrosome distance is altered, yet own and foreign are distinguished? The logic of why that allows suggesting that chromosomes are mixed in *cnb*-RNAi is not clear.

We updated the manuscript to explain that concept better. We emphasize that the proposed Cid measurements are simply an expression of how close NB and GMC Cid are to each other. This type of quantification allowed us to show that when MTOC activity is removed, Cid clusters behave significantly different from wild type hybrid cells. Because we increased the n's for both wild type and *cnb* RNAi hybrid cells, we are now able to conclude that MTOC activity is required to keep the two clusters separated until early metaphase when MTs from maturing centrosomes push Cid to the metaphase plate.

5) The study evokes “asymmetric chromatin-centrosome connections”. The connection between the chromosomes in the nucleus appears to be rather indirect (as the authors point out). Another interpretation: the apical centrosome and resulting microtubule network hold the nucleus in place and shape, which is perturbed if you lose the interphase MTOC. Consequently, in *cnb*-RNAi nuclear position and shape change and the nucleus might rotate (and perhaps changes in the distribution of other organelles) result. Can this be ruled out to contribute?

That is an interesting idea but ultimately still means that MTOC activity regulates Cid localization. So far, we have not seen any evidence that the nuclear envelope rotates. Also, if MTs simply hold the NE in place, then Cid should still remain in contact with the NE after colcemid treatment. That is not consistent with what we observe because Cid clusters appear to also occupy the center of the cell.

6) There are other ways to bring about loss of interphase MTOC activity in neuroblasts that could be tested.

We also used colcemid, *cnb* RNAi and *cnb* PACT.

7) The nuclear envelope analysis is descriptive and surprisingly not tested on *cnb*-RNAi fusion hybrids.

We imaged NE in wild type hybrid cells. In wild type hybrids, the NE becomes rather diffuse during mitosis (see Figure 6b; ~ 6:40 onward). For this reason, we considered generating *cnb* RNAi fusions not very informative.

Other

Does dual spindle formation require fused cells to be in the cell cycle? What about apical cells that withdraw from the cell cycle (according to FUCCI data) are they competent to form spindles when placed in a mitotic cytoplasm?

We generated hybrids of a GMC in G1-S, fused with a mitotic NB. The GMC quickly adjusted its cell cycle to the mitotic NB (Supplementary Figure 2g).

We assume apical cells would behave the same. However, we usually fuse basally located GMCs with the apical NB because cortex glia membranes surround NB and its GMC progeny. Given the stereotypic division axis of the NB (Loyer et al., 2018), GMCs are usually placed on the basal side and fusing these basally located GMCs with the apical NB is less intrusive and easier to perform.

Videos: Does cytokinesis take place normally upon fusion? None of the videos actually show that. This is important for the analysis of lagging chromosomes and the tumorigenicity test.

We have not seen cytokinesis failures. For instance, Supplementary Figure 1 shows two examples of successful NB-GMC fusions, imaged with *Sqh::GFP* (Myosin's regulatory subunit). In both cases we observe a robust cytokinetic ring forming and a midbody after ring closure. Similarly, we reanalyzed examples with lagging chromosomes, using *cherry::Jupiter* as an example and observe robust intercellular bridges forming (e.g Supplementary Figure 3c, e; Supplementary Figure 8).

Also, we removed the transplantation data from the manuscript.

Fig 6, this experiment, while interesting is not adding much. The sample size is likely to be too low to rule out that the fusion induces tumours. Have Brat controls been imaged and laser cut with unsuccessful fusion induction?

As mentioned above, we removed the transplantation data from the manuscript.

The quite specific configuration of neuroblasts centrosome regulation raise the question of transferability.

We disagree with this conclusion. First of all, male germline stem cells have been shown to also contain MTOC asymmetry. Second, our data here provide a new mechanism for chromatin retention and separation.

Fig 1 provide images of spindles resulting from fusions induced in interphase, prophase, metaphase...

This data is now shown in the new Supplementary Figure 3

Fig 2d provide images on how X and II actually look like

Data to that effect is shown in Supplementary Figure 3, 5, Supplementary Movie 5, Supplementary Movie 12.

Fig 4 the label is wrong this is Cid in blue not Histone 2A

Thanks. We corrected that.

Line 145 clarify to what "formed at the same time refers to"

We revised the manuscript and clarified this sentence.

Line 132 paragraph poses the question whether single or multiple spindles underly independent chromatin alignment at the beginning and finishes by concluding that spindle morphology is not important. Confusing, perhaps rewrite?

We have updated this paragraph.

Check methods some detail not visible.

The methods section was updated and contains additional experimental details (e.g Cid-cid distance measurements).

Line 168 "centrosome clustering" is an unlucky term. It suggests that centrosomes are found next to each other, while, if I got it right the fact that the NBs centrosomes are found in one spindle and the GMC centrosomes are found in the other is meant. Perhaps reword?

We changed the wording in the manuscript to 'positioning two centrosomes next to each other at each pole'

Information about the time resolution of imaging and when the cuts are made is critical, but missing for the experiments.

Time resolution is indicated in the revised methods. And the manuscript text usually mentions the time of the cuts. Alternatively, we specifically mention the timing of cutting in Supplementary Figure 3.

The sample size is rather small in places.

We increased sample number for wild type and *cnb* RNAi fusions (e.g Figure 2e,f; Figure 5h-j).

The criteria for choosing the site of cuts (I guess heterochromatin to judge apical) should be clearly stated and the site of cuts indicated in the relevant figures.

We updated supplementary Figure 1 accordingly. To save space, many of the figures start a few frames after the cut was done.

Reviewers' comments:

Reviewer #1 (Remarks to the Author):

It is an interesting piece of work with substantial new data and manuscript changes that increase the readability and the strength of the manuscript. I feel the study can be published after the authors respond to my last comments.

1-It is essential to state that the authors have created an artificial system. Possible defects could be induced by the ablation of membranes between the 2 NB and the GMC. To be fair, their chromatin separation mechanisms using the method described here may not apply to the first zygotic division in insects, arthropods, and vertebrates or potentially inform on the biased chromatid segregation in stem cells.

2-Paragraph L160 and discussion (L373). While the authors claim it is not the main point of this paper, it appears difficult to avoid previous work. In particular, I fail to see any difference in principle between the authors conclusion and the observations made in Rieder et al. study published in 1998. The results appear stickily identical. This is therefore fine to state clearly that 'the wait signal' principle described by the group of Sluder appear also valid in these NB-GMC fusions.

An important point: There is no clear mention of the 'anaphase onset signal' that propagates from the first to the neighboring spindle despite the attachment state of this "second" spindle. Please amend or be sure to use strong argument to demonstrate why this is different. Note that the distance effect may be less obvious in these cells.

3-Why is there only one MAD2-GFP spot (there are several kinetochores). Are they stuck together so they cannot be distinguished). Please amend or comment.

4-L192: a word is missing.

5-L335. "We conclude that nuclear envelopes establish a physical barrier between NB and GMC chromatin at least until metaphase in hybrid cells". This is at most a correlation. Unless the authors can demonstrate that the membranes surrounding the nuclei are really preventing fusion, I would suggest strongly moderating this somewhat over-assertive conclusion.

6-As the GFP-lamin signal decreases during mitosis, It would have been nice to show CD8-GFP instead of GFP-Lamin. To more strongly support their previous hypothesis (point 5).

7-Figure 6B: there is an error in side of the panels (tub+lamin, and Tub are inverted).

8-Discussion.Lane 367. "spindle realignment ensures that spindles co-align during metaphase" Please clarify.

Reviewer #2 (Remarks to the Author):

Authors have addressed most of the issues raised by this reviewer. Although the quality of some images remains poor and in many cases the dynamics of NB and GMC chromatin is not easy to interpret, weakening some of their conclusions, the manuscript is now acceptable for publication on Communications Biology.

Reviewer #3 (Remarks to the Author):

Overall, the manuscript has benefitted from the revision. The study has a range of relevant

findings and proposes interesting ideas. One limitation is that I still think there is a margin for error in the measurements of CID based on tracking fluorescent time points. For me there is a level of uncertainty how 'artificial' the experimental set up is in the sense that the transferability/comparability to other systems remains unsure, but since I did not raise this previously, it should not prevent publication. One aspect the authors may wish to consider is that there is an entire relevant literature that should perhaps be cited and discussed regarding the concept of "Rabl configuration" (Rabl. *Morphologisches Jahrbuch* 10, 1885). Neuroblasts clearly "Rabl". This concept encompasses the organisation of chromatin within the nucleus and importantly the clustering of centromeres to one site of the nuclear envelope as reported here. This configuration somehow links two cell divisions as the centromere positioning is thought to reflect the position of chromosomes in the preceding anaphase (e.g. Sexton et al *Cell* 148, 2012; Jin et al. *J. Cell. Biol.* 141, 1998). So there is a relevant well-known cell biological phenomenon/concept and the present work should be put into that context and explain how it does or does not align with this concept and what the advances regarding the relevance and function of the Rabl configuration potentially could be.

Response to reviewer's comments

Reviewer #1 (Remarks to the Author):

It is an interesting piece of work with substantial new data and manuscript changes that increase the readability and the strength of the manuscript. I feel the study can be published after the authors respond to my last comments.

1-It is essential to state that the authors have created an artificial system. Possible defects could be induced by the ablation of membranes between the 2 NB and the GMC. To be fair, their chromatin separation mechanisms using the method described here may not apply to the first zygotic division in insects, arthropods, and vertebrates or potentially inform on the biased chromatid segregation in stem cells.

We added the following statement to the discussion: "Nb-GMC fusions do not naturally occur and laser-based acute fusions could induce some unintended damage."

The system we use is artificial in the sense that fusions as presented here usually don't occur in vivo. However, in all fairness, cell culture in general and cell fusions with cultured cells is equally artificial.

I also think that based on the similarities between our system and the early mouse zygote – both systems display double spindles – we can gain mechanistic insight from our study that could be applicable to other systems.

2-Paragraph L160 and discussion (L373). While the authors claim it is not the main point of this paper, it appears difficult to avoid previous work. In particular, I fail to see any difference in principle between the authors conclusion and the observations made in Rieder et al. study published in 1998. The results appear stickily identical. This is therefore fine to state clearly that 'the wait signal' principle described by the group of Sluder appear also valid in these NB-GMC fusions.

An important point: There is no clear mention of the 'anaphase onset signal' that propagates from the first to the neighboring spindle despite the attachment state of this "second" spindle. Please amend or be sure to use strong argument to demonstrate why this is different. Note that the distance effect may be less obvious in these cells.

We rephrased the manuscript in the results section accordingly: "However, given the delays in GMC chromosome separation, we further conclude that ectopic spindles can initiate chromatid separation independently from the endogenous neuroblast spindle, perhaps because the diffusible wait anaphase signal is acting in a distance-dependent manner ⁸ or because the "start anaphase" signal overrides the inhibitory signal produced by unattached kinetochores ⁹."

3-Why is there only one MAD2-GFP spot (there are several kinetochores). Are they stuck together so they cannot be distinguished). Please amend or comment.

We don't know but some frames show several Mad2::GFP spots. We noticed that unfused wild type Nbs show a lot of unincorporated, diffuse signal, which could interfere with the resolution of individual spots.

4-L192: a word is missing.

Thanks. Fixed.

5-L335. "We conclude that nuclear envelopes establish a physical barrier between NB and GMC chromatin at least until metaphase in hybrid cells". This is at most a correlation. Unless the authors can demonstrate that the membranes surrounding the nuclei are really preventing fusion, I would suggest strongly moderating this somewhat over-assertive conclusion.

We rephrased this sentence as follows: These data suggest that nuclear envelopes establish a physical barrier between NB and GMC chromatin at least until metaphase in hybrid cells.

6-As the GFP-lamin signal decreases during mitosis, It would have been nice to show CD8-GFP instead of GFP-Lamin. To more strongly support their previous hypothesis (point 5).

We initially used CD8::GFP but because it labels multiple membranes, it was difficult to clearly see the nuclear envelope, especially in hybrid cells. For this reason, we used Lamin::GFP instead, which provided a much cleaner signal.

7-Figure 6B: there is an error in side of the panels (tub+lamin, and Tub are inverted).

Fixed. Thanks!

8-Discussion.Lane 367. "spindle realignment ensures that spindles co-align during metaphase" Please clarify.

We changed the sentence to "...centrosome migration and spindle realignment position both spindles next to each other during metaphase..."

Reviewer #2 (Remarks to the Author):

Authors have addressed most of the issues raised by this reviewer. Although the quality of some images remains poor and in many cases the dynamics of NB and GMC chromatin is not easy to interpret, weakening some of their conclusions, the manuscript is now acceptable for publication on Communications Biology.

Reviewer #3 (Remarks to the Author):

Overall, the manuscript has benefitted from the revision. The study has a range of relevant findings and proposes interesting ideas. One limitation is that I still think there is a margin for error in the measurements of CID based on tracking fluorescent time points. For me there is a level of uncertainty how 'artificial' the experimental set up is in the sense that the transferability/comparability to other systems remains unsure, but since I did not raise this previously, it should not prevent publication. One aspect the authors may wish to consider is that there is an entire relevant literature that should perhaps be cited and discussed regarding the concept of "Rabl

configuration” (Rabl. Morphologisches Jahrbuch 10, 1885). Neuroblasts clearly “Rabl”. This concept encompasses the organisation of chromatin within the nucleus and importantly the clustering of centromeres to one site of the nuclear envelope as reported here. This configuration somehow links two cell divisions as the centromere positioning is thought to reflect the position of chromosomes in the preceding anaphase (e.g. Sexton et al Cell 148, 2012; Jin et al. J. Cell. Biol. 141, 1998). So there is a relevant well-known cell biological phenomenon/concept and the present work should be put into that context and explain how it does or does not align with this concept and what the advances regarding the relevance and function of the Rabl configuration potentially could be.

We thank the reviewer for this helpful suggestion. We added a paragraph to the discussion to take the Rabl configuration concept into account.

REVIEWERS' COMMENTS:

Reviewer #1 (Remarks to the Author):

The authors have clarified my concerns. The manuscript is now suitable for Communication Biology.

Reviewer #3 (Remarks to the Author):

I think the manuscript reads well now.